# SHORTCUTSBENCH: A LARGE-SCALE REAL-WORLD BENCHMARK FOR API-BASED AGENTS

**Haiyang Shen[1], Yue Li[3], Desong Meng[4], Dongqi Cai[5], Sheng Qi[2], Li Zhang[5], Mengwei Xu[5], Yun Ma[1]***

[1]Institute for Artificial Intelligence, Peking University
[2]School of Computer Science, Peking University
[3]School of Software & Microelectronics, Peking University
[4]School of Electronics Engineering and Computer Science, Peking University
[5]Beijing University of Posts and Telecommunications
{hyshen}@stu.pku.edu.cn, {mayun}@pku.edu.cn✉

## ABSTRACT

Recent advancements in integrating large language models (LLMs) with application programming interfaces (APIs) have gained significant interest in both academia and industry. Recent work demonstrates that these API-based agents exhibit relatively strong autonomy and planning capabilities. However, their ability to handle multi-dimensional difficulty levels, diverse task types, and real-world demands remains unknown. In this paper, we introduce SHORTCUTSBENCH, a large-scale benchmark for the comprehensive evaluation of API-based agents in solving real-world complex tasks. SHORTCUTSBENCH includes a wealth of real APIs from Apple Inc., refined user queries, human-annotated high-quality action sequences, detailed parameter filling values, and parameters requesting necessary input from the system or user. We revealed how existing benchmarks / datasets struggle to accommodate the advanced reasoning capabilities of existing more intelligent LLMs. Moreover, our extensive evaluation of agents built with 5 leading open-source (size $\geq$ 57B) and 5 closed-source LLMs (e.g. Gemini-1.5-Pro and GPT-4o-mini) with varying intelligence level reveals significant limitations of existing API-based agents in the whole process of handling complex queries related to API selection, parameter filling, and requesting necessary input from the system and the user. These findings highlight the great challenges that API-based agents face in effectively fulfilling real and complex user queries. All datasets, code, experimental logs, and results are available at https://github.com/EachSheep/ShortcutsBench.

## 1 INTRODUCTION

Large language model based agents (LLM-based agents) (Zhao et al., 2025; Chen et al., 2025; Wang et al., 2024b; Xi et al., 2023) built on application programming interfaces (APIs) have gained significant interest in academia and industry. By integrating LLM with APIs, LLMs can access real-time information (Qin et al., 2024), reduce hallucination with external knowledge (Gao et al., 2024), as well as plan and complete complex tasks that need multi-step actions (Gravitas, 2024). Many of these agents (OpenAI, 2024a) have already demonstrated commendable performance on simple

Table 1: Less intelligent LLMs (even $3B$) on existing benchmarks / dataset demonstrated excellent results with the same prompt in Section 4.1.

| Acc. (%) | MetaTool 2024b | ToolLLM 2024 | ToolBench 2024 |
|---|---|---|---|
| **LLaMA-3.2-3B** | 89.64 | 72.92 | 79.47 |
| **QWen-2.5-3B** | 88.29 | 77.86 | 91.35 |
| **LLaMA-3-8B** | 89.00 | 78.31 | 93.57 |
| **QWen-2.5-7B** | 92.50 | 82.69 | 94.26 |
| **GPT-4o-mini** | 88.31 | 84.50 | 89.90 |

tasks involving only a few actions such as "*Check the weather ① and tell me ②*". This impressive performance raises an important question: Are these API-based agents truly capable of generating action sequences for real and complex demands?

---
*Corresponding author

Some existing benchmarks / datasets[1] have attempted to evaluate API-based agents. However, they have three limitations (please refer to **Table 2 for all details**): **First**, the APIs (a.k.a tools available to the agent) lack richness, and the queries (a.k.a the task to the agent) lack complexity. They either involve a limited number of APIs, cover small numbers of apps (an app may have $\geq 1$ APIs), or the difficulties of the queries are limited in a narrow range, with the average action length ranges from 1 to 5.9. This lack of richness and complexity makes it difficult to effectively distinguish the capabilities of different agents, even on less intelligent LLMs like `QWen-2.5-3B` (Alibaba, 2024), let alone more intelligent LLMs like `Gemini-1.5-Pro` (Google, 2024). Our evaluation on API selection of these less intelligent LLMs on 3 representative[2] benchmarks / datasets (Table 1) shows that even 3B LLMs can achieve impressive results. There is almost no difference in accuracy across LLMs of varying intelligence levels. Therefore, existing benchmarks / datasets struggle to accommodate more intelligent LLMs and to differentiate the intelligence levels among various LLMs. **Second**, the APIs lack realism as they may be manually crafted, and the queries fail to reflect actual user demands since they may be either created by hand or generated directly by LLMs without verifying real user demands. Moreover, they only cover the evaluation of API selection, lacking a study on API parameter filling. Efficient and accurate parameter filling is essential for an agent to finish the whole process of completing queriess **Third**, they don't adequately evaluate the agent's ability to request systems or the users for the necessary input to resolve the missing information for solving the queries. This is crucial as a user's query may be implicit or may not provide all the input an agent needs to solve the task effectively.

In this paper, we innovatively propose to use data extracted from existing *Digital Automation Platforms* (DAPs), *Apple Shortcuts*, to construct a high-quality benchmark for API-based agents, i.e., SHORTCUTSBENCH. To the best of our knowledge, SHORTCUTSBENCH is the first large-scale real API-based agent benchmark considering APIs, queries, and action sequences. SHORTCUTSBENCH provides rich and real APIs, queries with various difficulties and task types, high-quality human-annotated action sequences, and queries from real user demands. Moreover, it also provides precise values for parameter filling, including primitive data types, enum types, and the use of output from previous actions for parameter values, as well as evaluations of the agent's awareness in requesting necessary input from the system or user. Furthermore, the scale of APIs, queries, and the corresponding action sequences is comparable or even better to benchmarks / datasets created by LLM or modified by existing datasets. The overall comparison between SHORTCUTSBENCH and existing benchmarks / datasets is listed in Table 2.

To demonstrate SHORTCUTSBENCH's advantages, we do extensive evaluations of API-based agents from 10 leading open-source and close-source LLMs, covering varying intelligence levels. To our best known, this is the most comprehensive evaluation considering the API selection, parameter value filling, and recognition of the need for input from the system or the user, covering all key processes of API-based agent. The evaluation results highlight great limitations of existing API-based agents.

In summary, this paper makes the following key contributions:

- We identified problems of the existing benchmarks / datasets, specifically that they struggle to accommodate the advanced reasoning capabilities of existing more intelligent LLMs, and have conducted experiments to validate the problem.
- We innovatively extracted data from Shortcuts, to build a high-quality benchmark for API-based agents. To our best knowledge, SHORTCUTSBENCH is the most realistic, rich, comprehensive, and large-scale benchmark for API-based agents. We hope this approach to dataset construction will inspire more researchers.
- We made efforts to evaluate 10 advanced LLM-based agents with varying intelligence levels on the whole process required to complete user queries, including API selection, parameter filling, and their awareness of requesting necessary input from the system or user.
- We obtained massive interesting conclusions such as (1) Open-source LLM agents now match closed-source ones on simpler tasks but still lag behind on complex ones; (2) Extracting necessary parameters from queries is the most challenging task in parameter filling; (3) There is a substantial lack of awareness in agents when it comes to requesting the necessary input;

---

[1] We refer to the evaluation-specific datasets as "benchmarks" and fine-tuning datasets as "datasets".

[2] MetaTool uses the native GPT API, while ToolBench and ToolLLM have the longest average action length and the largest scale with real-world API, respectively.

Table 2: SHORTCUTSBENCH has a great advantage in the ①realness and richness, ②the complexity of APIs, queries, and corresponding action sequences, ③the validity of action sequences, ④detailed parameter value filling, ⑤the awareness for asking necessary input, and ⑥the overall scale.

| Resource | Shortcuts Bench (Ours) | Meta Tool 2024b | Tool LLM 2024 | API Bench 2024 | Tool Alpaca 2023 | API Bank 2023 | Tool Bench 2024 | Tool QA 2024 | Tool Lens 2024 |
|---|---|---|---|---|---|---|---|---|---|
| **Real API?** | ✓ | ✓ | ✓ | ✓ | ✓ | ✗ | ✗ | ✗ | ✓ |
| **Demand-driven Query?** | ✓ | ✗ | ✗ | ✗ | ✗ | ✗ | ✗ | ✗ | ✗ |
| **Human-Annotated Act.?** | ✓ | ✗ | ✗ | ✗ | ✗ | ✗ | ✗ | ✗ | ✗ |
| **Multi-APIs Query?** | ✓ | ✓ | ✓ | ✗ | ✗ | ✗ | ✗ | ✓ | ✓ |
| **Multi-Step Act.?** | ✓ | ✓ | ✓ | ✗ | ✓ | ✓ | ✓ | ✓ | ✓ |
| **Prec. Val. for Para. Fill?** | ✓ | ✗ | ✗ | ✗ | ✗ | ✗ | ✗ | ✗ | ✗ |
| **Awareness for Ask Info?** | ✓ | ✗ | ✗ | ✗ | ✗ | ✗ | ✗ | ✗ | ✗ |
| **# Apps** | 88 | N/A | 3451 | 3 | N/A | N/A | 8 | N/A | N/A |
| **# APIs** | 1414 | 390 | 16464 | 1645 | 53 | 400 | 232 | 13 | 464 |
| **# Queries** | 7627 | 21112 | 12657 | 17002 | 3938 | 274 | 2726 | 1530 | 18770 |
| **# Avg APIs** | 9.62 | 1.02 | 2.3 | 1.0 | 1.0 | 2.1 | 5.4 | 3.5[*] | 2.65 |
| **# Avg Actions** | 21.62 | 1.02 | 4.0 | 1.0 | 1.0 | 2.2 | 5.9 | 3.9[*] | 2.67 |

[*] denotes estimation.

- We have fully open-sourced all the datasets, code, experimental logs, and results, and provided detailed documents. We hope our research opens new directions for the real-world deployment of existing LLM-based agents.

## 2 RELATED WORK

**API-based agents.** API-based agents treat APIs as tools. They accept queries, generate action sequences based on queries and provided APIs, and generate next action depends on the history actions (Wang et al., 2024b; Yao et al., 2023). Related work about API-based agents can generally be categorized into 3 types: (1) Task-specific enhancement focuses on improving the agent's ability like using the model (Shen et al., 2024; Zhong et al., 2024). (2) Data-driven workflows emphasize the importance of data by researching how to construct action sequences, enabling generated data to fine-tune the model (Qin et al., 2024; Patil et al., 2024). (3) Agent evaluation studies the assessment of agents (Huang et al., 2024b; Li et al., 2023).

**Code-based agents.** Code-based agents use code generated for interaction with the external environment. They accept queries, generate scripts in programming languages such as Python (OpenAI, 2024b; Wang et al., 2024c), JavaScript (Wang et al., 2024a; Zheng et al., 2023), or Shell (OpenInterpreter, 2024; Sladić et al., 2024), and then input the code into interpreters. The execution results are then returned to the agent, which is used to help determine the next code generation. Currently, these approaches primarily focus on enhancing agent performance in specific tasks by incorporating additional knowledge (Wang et al., 2024a; Wu et al., 2023), increasing feedback (OpenInterpreter, 2024; Huang et al., 2024a), and decomposing tasks (Huang et al., 2023; Prasad et al., 2024). In addition to work on optimization methods, numerous efforts have emerged to evaluate code-based agents (Trivedi et al., 2024; Liu et al., 2024)

**Digital Automation Platforms (DAPs).** DAPs (Abdou et al., 2021) refer to software tools or services designed to optimize workflows through automation. DAPs leverage technologies such as robotic process automation (RPA) (Chakraborti et al., 2020) and low-code / no-code development tools to achieve the goals. DAPs like *Zapier* (Zapier, 2024), *Make* (Make, 2024), and *IFTTT* (Rahmati et al., 2017) offer extensive APIs that enable users to create automated workflows. Similarly, DAPs such as *Microsoft Power Automate* (Microsoft, 2024) and *Tasker* (Dias, 2024) are primarily used to build workflows on *Azure* and *Android*, respectively. Recently, with the rise of LLM-based agents, platforms like *Coze* (Coze, 2024) and *Dify* (Dify, 2024) have emerged as "agent construction

platforms". Functionality like "workflow" in these platforms can also help manually build workflows, but they have been specifically optimized for integration with LLMs.

*Shortcuts app* (formerly *Workflow*) (Apple, 2024) is an app developed by Apple for building workflows through a graphical interface, available on Apple's operating systems (iOS / iPadOS and macOS). Shortcuts app can be seen as the DAP of Apple. It allows users to create workflows (known as *shortcuts* (Apple, 2024c)) that execute specific tasks on their devices and share them online via iCloud link (Apple, 2024b). Users can also download curated shortcuts from the *Gallery* of the Shortcuts app. However, the shortcuts available in the Gallery are very limited, with only a few dozen options. To access more shortcuts, users must either collect them from third-party sharing sites like *Shortcuts Gallery* (Gallery, 2024) or create their own. Shortcuts can be triggered through the Shortcuts app, widgets, the share sheet, old Siri (Apple, 2024a), new Siri of Apple Intelligence (Apple, 2024b), and they can also be automated to run upon specific events.

Shortcuts are composed of multiple API calls (actions). An agent can use the shortcut as a whole API or utilize the individual APIs involved in the shortcut. This paper treats APIs within the shortcuts as APIs available to the agent, aiming for the agent to automatically construct workflows of API calls.

## 3 DATASET

In this Section, we first introduce the acquisition of the dataset (Section 3.1). Then, we outline the SHORTCUTSBENCH's construction process (Section 3.2). Finally, we outline the setup for evaluation tasks to evaluate the agent's ability to handle tasks of varying difficulty, including the ability to select suitable APIs (Section 3.3.1), the ability to do parameter filling (Section 3.3.2), and the awareness in requesting additional input from the system or user (Section 3.3.3).

### 3.1 DATASET ACQUISITION

Figure 1 shows the data acquisition process. We first use search engines to identify popular public shortcut-sharing sites ①. We totally find 14 sites (Table 5). Then we crawled these sites to obtain fields such as "shortcut name", "function description", "shortcut type", and "iCloud link" ②. Then we downloaded the shortcut source file by "iCloud link" and then perform deduplicating based on iCloud link itself and the actual shortcut content (i.e., the action sequences) ③. Subsequently, we extracted "app name" using the field `WFWorkflowActionIdentifier` in the shortcut source file, and then downloaded associated apps ④. These apps may come from various sources. (1) apps from the macOS or iOS App Store, (2) apps like *Keynote* from path `/Applications/` and `/System/Application/` on macOS, (3) third-party apps from the the official website of the app. During the downloading, we also excluded some legacy apps and paid apps.

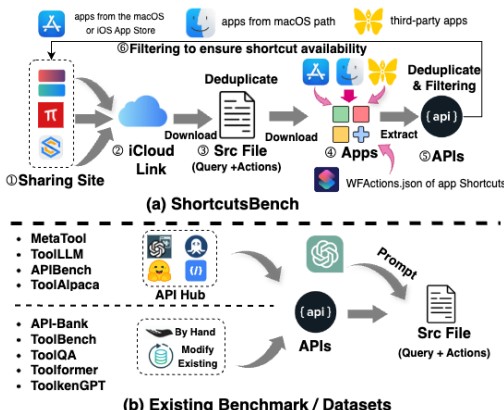

Figure 1: (a) illustrates the data acquisition process. (b) shows the dataset acquisition of existing work. APIs in existing work are collected from API hubs, created by hand, or modified from existing datasets. The queries and action sequences are constructed using templates or semi / fully automated methods.

Then we managed to extract APIs from the downloaded apps ⑤. The APIs are mainly from intent definition file `${filename}.actionsdata` from *AppIntent* (Apple-Inc., 2024b) framework and `${filename}.intentdefinition` from *SiriKit* (Apple-Inc., 2024c) framework. We extracted all APIs involved in the apps. During the extraction, we perform deduplication of APIs based on manually crafted rules as an app may have multiple duplicate API definition files with the same API definition. This process also involves significant manual filtering. Additionally, for app *Shortcuts*, which are deeply integrated with Apple's operating system, we need to obtain their API definition files `WFActions.json` from system path `/System/Library/PrivateFrameworks/WorkflowKit.framework/` on macOS, instead of extracting it from the app itself. Subsequently,

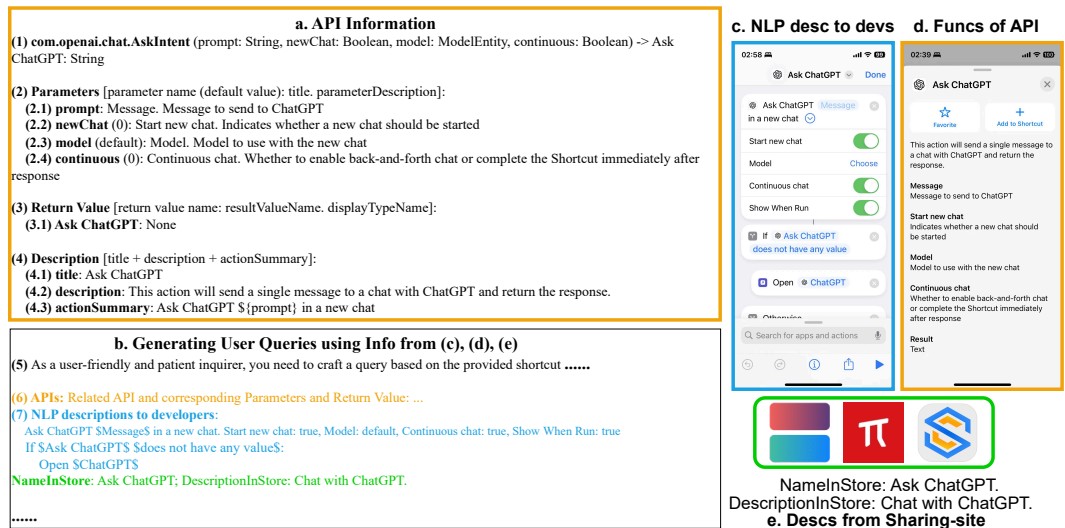

Figure 2: The construction of SHORTCUTSBENCH. (a) shows the information of API `com.openai.chat.AskIntent` extracted from the app ChatGPT's `${filename}.actionsdata`. We provide this API description to the LLM, expecting it to call the API at the appropriate time. The API information shown in (a) includes the API functionality description (a.k. (a.1)~(a.4)) as shown in (d), and the user-friendly natural language description of the API (a.k. (a.4.3)) seen by shortcut developers during programming, as shown in (c). (e) presents the shortcut name and functionality description from the shortcut sharing-sites. (b) shows the simplified prompt fed to `GPT-4o`, instructing it to generating queries based on demands indicated by shortcuts by integrating the info from (c), (d), and (e). Different colors indicate different information sources.

we further filtered the shortcuts ⑥ based on criteria such as whether the associated apps is paid app, whether the apps were outdated, and whether the APIs were deprecated. Additionally, we imported all shortcuts into the macOS Shortcuts app to ensure they were functional. Finally, as shown in Table 2, we get 88 apps from various categories such as "Health & Fitness" and "Developer Tools". We finally get 1414 APIs involved in 7627 shortcuts.

As the acquisition process involves specific knowledge about shortcut-sharing sites and Apple's operating system, detailed explanations are omitted here due to space constraints. For more details about the whole acquisition process, please refer to Appendix A.1.

## 3.2 DATASET CONSTRUCTION

As shown in Figure 2, existing benchmarks / datasets consist of two parts: (1) APIs; (2) queries and corresponding action sequences.

**APIs** (Figure 2.a) include the "API description" (a.4), "API name" (a.1), "parameter names" (a.2), "parameter types" (a.1), "default value" (a.2), "return value type" (a.3), and "return value name" (a.3). The field names in square brackets `[]` represent the original field name in the shortcut source file. For more details about `${filename}.actionsdata`, `${filename}.intentdefinition`, and `WFActions.json`, please refer to the Appendix A.2. In existing benchmarks / datasets, the "parameter types" and "return value types" are composed of primitive data types such as `int` and `string`. In addition to primitive data types, APIs in SHORTCUTSBENCH also include "enum" or "advanced data types". Enum is composed of "the class name" and "the possible value", with each value equipping a "value name". We also provide the agent with a description of the "enum" in the API information. Advanced data types, such as the `model` (a.1), include three `String` types named `identifier`, `title`, and `subtitle`. We can comprehend them through their "type name" and "type description".

**Query and action sequence.** A *query* is a user command, such as "*Tell me what the weather will be like tomorrow.*" The *action sequence* (aka. shortcut) is the series of API calls to complete the

query, with each API call referred to as an *action*. The action sequence identifies the steps needed to complete a query. As shown in Figure 1.b, existing benchmarks / datasets collect APIs first and then use them, either fully automatically or semi-automatically, to construct query and action sequences through LLMs. In contrast, action sequences in SHORTCUTSBENCH are all human-annotated. The shortcut developers are our annotators. APIs in SHORTCUTSBENCH are also all real-world. Moreover, we ensured the quality of action sequences by filtering shortcuts based on criteria such as whether the associated apps were paid, outdated, or relied on deprecated APIs. We also imported all shortcuts into the macOS Shortcuts app to verify their functionality.

*Generating queries.* As shown in Figure 1, existing works construct query and action sequences based on available APIs. In contrast, we construct queries based on existing action sequences and APIs. When constructing a query for a specific action sequence, we need to understand the functional description of the action sequence (Figure 2.e) and detailed information about the involved APIs (Figure 2.a). With this information, we can generate higher-quality queries. To ensure the quality of the generated queries, we also leverage the unique advantage of shortcuts: the natural language workflow descriptions (Figure 2.b.7 / Figure 2.c). By inputting these intuitive natural language descriptions into an LLM, we can generate more accurate queries. When generating queries, we also require the model to naturally include primitive data type parameters and enum data type needed for API calls in generated queries. This helps us evaluate the agent's ability to fill in primitive parameters in Section 3.3.2. To ensure the quality of generated queries, we use the state-of-the-art LLM, `GPT-4o` (OpenAI, 2024), to generate the queries. The prompt templates we used to generate queries can be found in the Appendix A.2.

To ensure the quality of queries generated, follow existing work (Qin et al., 2024), we conducted a preliminary experiment using 3 LLMs: `GPT-4o`, `GPT-3.5`, and `Gemini-1.5-Pro`, on a dataset of 100 samples. The results showed that human evaluators rated `GPT-4o` generated queries the highest, outperforming the other 2 LLMs. `GPT-4o` demonstrated superior performance by accurately capturing required parameters and providing clear query descriptions, meeting our criteria in 94/100. This superior performance can largely be attributed to the natural language workflow descriptions.

## 3.3 TASK DEFINITION AND METRICS

We aim to address 3 research questions regarding the performance of existing agents built using leading LLMs on SHORTCUTSBENCH with varying difficulties: **(1)** How do they perform in API selection? **(2)** How do they handle API parameter value filling, including parameters for primitive data types, enums, and outputs from previous actions? **(3)** Can they recognize when input is required for tasks that need system or user input?

**Preliminaries.** SHORTCUTSBENCH consists of a set of queries $Q = \{q_1, q_2, ..., q_n\}$, corresponding "golden" action sequences $ASeq = \{aseq_1, aseq_2, ..., aseq_n\}$, and all available APIs $APIs = \{api_1, api_2, ..., api_m\}$. For each query $q_i, 1 \leq i \leq n$, the corresponding "golden" action sequence is $aseq_i = \{a_1, a_2, ..., a_{|aseq_i|}\}$, where the length of the

Table 3: Final evaluation set with varying difficulties.

| $|aseq_i|$ | (0, 1] | (1, 5] | (5,15] | (15,30] | Overall |
|---|---|---|---|---|---|
| **# Queries** | 706 | 2169 | 1571 | 774 | 5220 |
| **# Avg APIs** | 1.17 | 3.43 | 8.30 | 13.76 | 6.60 |
| **# Avg Acts** | 1.00 | 3.19 | 9.60 | 21.58 | 8.34 |

action sequence is $|aseq_i|$. Each app $app_j$ has a set of APIs $apis_j = \{api_1, api_2, ..., api_{|apis_j|}\}$. The action sequences generated by the agent for each query $q_i$ are referred to as $bseq_i$.

**Prepare available APIs for each query.** For each query $q_i$, we provide the LLM with a certain number of usable APIs to simulate real-world scenarios where APIs can be input into the LLM's context. Following existing work (Qin et al., 2024; Tang et al., 2023; Xu et al., 2024), we equip each $q_i$ with a specific number of APIs. For each $aseq_i$, let $|APIs_i|$ represent the number of APIs involved. In addition to these $|APIs_i|$ APIs, we equip each query with extra APIs calculated as $max(min(x \times |APIs_i|, 20 - |APIs_i|), 0)$, where $x \in \{3, 4, 5\}$. We do this because it is impractical to input all APIs into the context simultaneously. When dealing with a large number of APIs, additional retrieval is often required (Qin et al., 2024; Qu et al., 2024), which we do not consider in this work.

**Further Processing.** Considering the context limitations of LLMs, we excluded shortcuts longer than 30 and parts using the API `is.workflow.actions.runworkflow` to call other shortcuts.

While these shortcuts remain in our open-source dataset, they will not be included in the subsequent evaluation. We aim to study the performance of agents on queries of varying difficulties. As shown in Table 3, we categorize SHORTCUTSBENCH into 4 difficulty levels and 8 task types based on $|aseq_i|$ and "shortcut type", respectively. For more details, please refer to the Appendix A.3. When calculating the length, for branching actions like `is.workflow.actions.conditional`, we consider the longest branch as the length. Additionally, we ignore the lengths of looping actions like `is.workflow.actions.repeat.count` and special actions such as `is.workflow.actions.comment`. Due to the presence of branching actions, the average number of APIs involved when $p = 1$ is greater than one, specifically 1.17. For a detailed process, please refer to the Appendix A.3. The number of shortcuts in each level is denoted as $n_p$. Each query and action sequence is referred to as $q_{p,i}$ and $aseq_{p,i}$, with $1 \leq p \leq 4$ and $1 \leq i \leq n_p$.

### 3.3.1 PERFORMANCE ABOUT API SELECTION

Following existing work (Huang et al., 2024b; Patil et al., 2024; Xu et al., 2024), we use the accuracy of API selection as the metric. The accuracy is calculated as the number of correct API selections $m_p$ divided by $n_p$. Specifically, each time we predict an action $b_j, 1 \leq j \leq |aseq_i|$, we provide the agent with all the correct historical actions $\{a_1, a_2, ..., a_{j-1}\}$. We then require the agent to predict the next action. All actions predicted by the agent form the prediction sequence $bseq_{p,i}$. This method is similar to the next token prediction (NTP) in LLMs, effectively preventing a cascade of errors in subsequent action predictions due to a single incorrect prediction. During the prediction, when encountering special actions such as branching and looping, we skip predicting these actions and directly add them to the historical actions. For more details, please refer to Appendix A.4. We chose API selection accuracy over the final result for the following two additional reasons:

- SHORTCUTSBENCH contains numerous APIs such as *opening the "All Shortcuts Folder" in the Shortcuts app* that do not have a return value. This makes it challenging to evaluate using existing metrics that measure the success rate of solving queries (Qin et al., 2024; Xu et al., 2024;?).

- SHORTCUTSBENCH includes numerous APIs with complex input and output types, such as `PDFs` and `Rich Text`. Converting these formats into text that an LLM can process presents a significant challenge (Naveed et al., 2023), as LLMs struggle to serialize them into text. Consequently, it becomes difficult to ascertain the correctness of the final results. However, measuring API selection accuracy is straightforward.

### 3.3.2 EFFECTIVENESS OF API PARAMETER VALUE FILLING

In this part, we aim to investigate the performance of agents in API parameter value filling, including parameters for "primitive data types" and "enums" and filling output from previous actions. For each input parameter of every action in SHORTCUTSBENCH, we expect the agent to fill in the following parameters correctly:

- **Static Parameters Preset:** These are static parameters that users provide as default inputs of the action. These static parameters typically include primitive data types such as `String` and `Integer`, as well as custom `Enum` defined by app developers. When the query explicitly specifies a parameter that can be used as a static parameter, we expect the agent to accurately fill in the parameter values according to the user's query and the API's definition. When generating queries, we have already required the LLM to naturally include primitive and enumerated data types (Section 3.2). To further ensure that the corresponding parameters are indeed included in the queries during evaluation, we used the LLM to filter these parameters further, ensuring their presence in the queries. Detailed prompts can be found in the Appendix A.5.

- **Outputs from Previous Actions:** An action may either have no output or, if it does have an output, the output may be used by the following actions. In SHORTCUTSBENCH, outputs that are difficult to input directly into the LLM are represented by a unique identifier (`UID`) and an output name (`OutputName`), which can be input into the LLM for processing. The agent should have the ability to correctly use the output values of previous actions.

For the static parameters preset, we evaluate using the overall parameter fill rate. Let $sppa_i$ be the total number of parameters that need to be filled in $aseq_i, 1 \leq i \leq n_q$, where $n_q$ is the number of queries. If the agent correctly fills $sppt_i$ parameters in the generated action sequence $bseq_i$, then

the **s**tatic **p**arameter **p**reset accuracy can be calculated as $Acc_{spp} = \sum_{i=1}^{n_q} sppt_i / \sum_{i=1}^{n_q} sppa_i$. Similarly, for **o**utputs **f**rom **p**revious **a**ctions, the accuracy can be calculated as $Acc_{ofpa} = \sum_{i=1}^{n_q} ofpat_i / \sum_{i=1}^{n_q} ofpaa_i$.

### 3.3.3 RECOGNITION OF NEED FOR INPUT

In this section, we aim to investigate the ability of existing API-based agents to ask systems or users for necessary input to resolve the missing information. This missing information can come from the system like clipboard (`Clipboard`), input files (`ExtensionInput`), and the current date (`CurrentDate`) or from the user (`Ask`) (Apple-Inc., 2024a). For example, a parameter named `tags` is usually represented in a shortcut as `"tags":{"Value":{"Type": "Ask"}}`, where `"Type": "Ask"` indicates that the parameter will prompt the user for input. For more details, please refer to Appendix A.6. We use the proportion of correctly identified parameters to evaluate the agent's ability to recognize the need for input from the system or the user. Let $n_s$ be the number of queries, $aska_i$, $askt_i$ be the number of times the need from the system or the user appears in $aseq_i$, $bseq_i$, respectively, The accuracy of **a**sk **f**or **n**ecessary **i**nformation can be calculated as $Acc_{afni} = askt_i / aska_i$.

## 4 EVALUATION

### 4.1 SETUP

**Model.** Referencing existing work (Huang et al., 2024b; Qin et al., 2024; Li et al., 2023), considering the performance of existing LLMs, we selected 10 most advanced LLMs to construct API-based agent. The chosen model includes 5 closed-sourced and 5 open-source LLMs, covering varying intelligence levels. Among them, `Gemini-1.5-Pro`, `LLaMA-3-70B`, `QWen-2-70B`, and `Deepseek-2-chat/coder` are LLMs benchmarked against `GPT-4o-2024-05`, while `Gemini-1.5-Flash`, `ChatGLM-4-Air`, and `QWen-2-57B` are benchmarked against `GPT-4o-mini-2024-07` and `GPT-3.5-turbo`. We did not evaluate `GPT-o1-preview/mini`, `GPT-4o`, `LLaMA-3.1-70b/405B`, and `QWen-2.5-72b`, mainly due to limited access, high costs, and the fact that LLaMA and QWen are minor version improvements with recent releases.

We also compare our work with specialized API-calling fine-tuned LLMs, such as AgentLM (Zeng et al., 2024) and xLAM (Zhang et al., 2024), uncovering intriguing findings. Due to space limitations, the results are presented in Appendix A.9 for interested readers.

**Prompt Template.** Following existing work (Huang et al., 2024b; Qin et al., 2024; Tang et al., 2023; Zhuang et al., 2024), we slightly modified the ReACT (Yao et al., 2023) templates to construct the API-based agents. For all 3 research questions (RQs), we use the same prompt templates. An agent should correctly select APIs, fill in parameters, and be aware of the need to request necessary input from the system or user at appropriate times. Please refer to Appendix A.7 for more details.

### 4.2 RESULT ANALYSIS

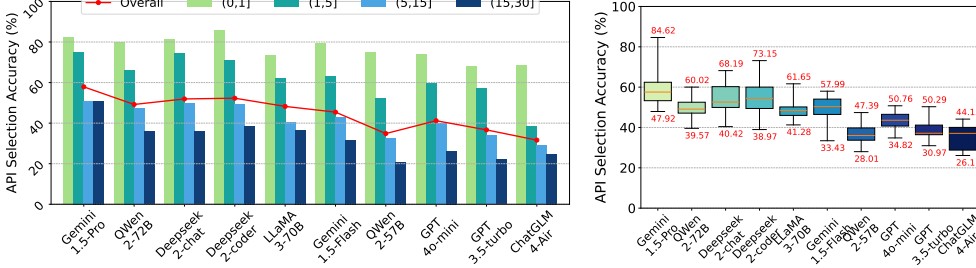

Figure 3: The API selection accuracy on queries with different complexity levels.

Figure 4: The API selection accuracy difference of each LLM across 8 task types.

From Figure 3, we can see that for tasks with a lower difficulty level, both less intelligent LLMs and more intelligent LLMs perform well. This is similar to the conclusion drawn from Table 1. However,

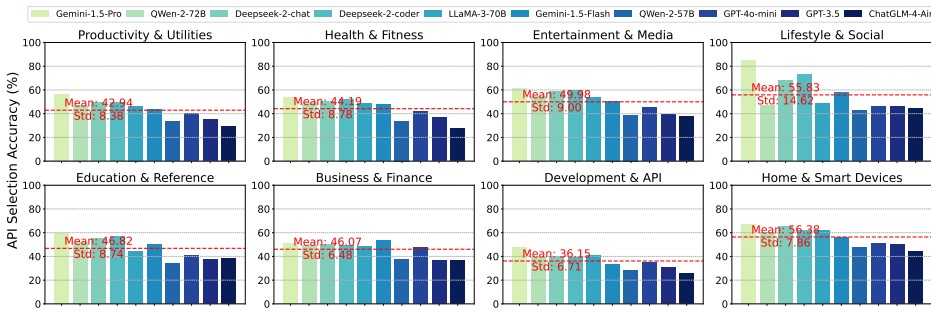

Figure 5: The API selection accuracy of each task type on 10 API-based agents.

for tasks with a higher difficulty level, only the more intelligent LLMs like `Gemini-1.5-Pro`, `Deepseek`, and `QWen2` perform adequately.

**The superiority of SHORTCUTSBENCH.** Combined with Table 1, SHORTCUTSBENCH can effectively distinguishes between different levels of intelligence, making it a superior benchmark.

Through the results of API selection accuracy (Section 3.3.1), we get the following conclusions:

- **Agents built using open-source LLMs now perform comparably to closed-source LLMs on lower-difficulty tasks but still lag on higher-difficulty tasks.** From Figure 3 we know that open-source LLMs $>=$ 70B match the performance of closed-source LLMs from the first 3 difficulty tasks, significantly outperforming `GPT-4o-mini` and `GPT-3.5-turbo`. However, they still lag behind closed-source LLMs in handling complex tasks at the 4-th level. For more details, please refer to Appendix A.8.

- **Existing LLM-based agents still perform poorly on tasks requiring multi-step reasoning, even more intelligent LLMs like `Gemini-1.5-Pro` struggle with high-difficulty tasks.** From Figure 3 we know that almost all LLMs handle well in API selection tasks at the level of (0,1], but only more advanced models like `Gemini-1.5-Pro` and `QWen-2-72B` can do well in higher-difficulty tasks of (1,5]. As tasks become more complex, the accuracy drops sharply. The average accuracy dropped by $19\%$ as task difficulty rose from (0,1] to (1,5], ranging from a $9\%$ decrease (`Deepseek-2-chat`) to a $44\%$ (`ChatGLM-4-Air`). From (0,1] to (5,15], accuracy fell by $46\%$, with drops from $38\%$ (`Gemini-1.5-Pro`) to $58\%$ (`ChatGLM-4-Air`).

- **Agents built with the same LLM show significant performance variations across different types of tasks.** From Figure 5 we know that the performance difference of agents built with different LLM ranges from $15.94\%$ (`GPT-4o-mini`) to $36.70\%$ (`Gemini-1.5-Pro`).

- **Existing API-based agents perform well on tasks in daily life such as Lifestyle & Social but show poorer performance on professional tasks like Development & API.** From Figure 5 we know that `Lifestyle & Social` exhibit the highest average accuracy, surpassing the lowest category, `Development & API` by approximately $18\%$.

Based on the results of API Parameter Value Filling (Section 3.3.2), we draw following conclusions:

- **API selection and parameter filling both impact the agent's performance. However, API selection has a greater effect.** As shown in Figure 6a, for existing more intelligent LLM like `Gemini-1.5-Pro`, increased task difficulty has a much smaller impact on the accuracy of parameter filling, especially on using outputs from previous actions. This indicates that the greatest limitation of existing API-based agents in addressing user queries lies in the reasoning and planning capabilities implied by API selection.

- **The performance of API parameter filling remains a bottleneck for existing less intelligent LLMs.** As shown in Figure 6a, the performance of less intelligent LLMs like `GPT-4o-mini` in API parameter filling significantly decreases as task difficulty increases.

- **Compared to using the outputs of previous actions, extracting relevant parameters from the user's query and filling them is more challenging.** As shown in Figure 6a, the colors in the top plot (filling primitive data types and enum data types) are generally lighter than those in the

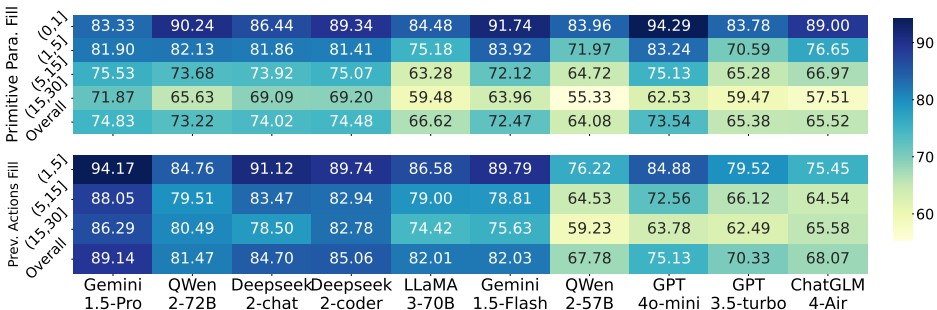

(a) Accuracy of primitive data types & enum data types (upper) and outputs from previous actions (lower).

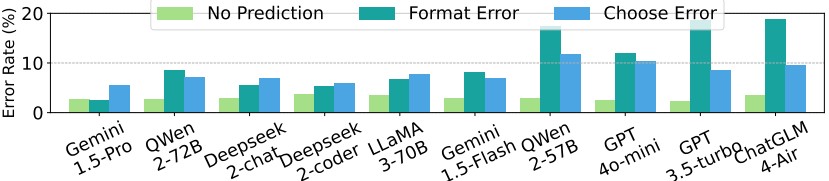

(b) The error rates for action parameter value filling.

Table 4: The accuracy of recognition of the need for input from the system or the user.

| Levels | Gemini 1.5 Pro | QWen 2 72B | Deep seek2 chat | Deep seek2 coder | LLaMA 3 70B | Gemini 1.5 Flash | QWen 2 57B | GPT 4o mini | GPT 3.5 turbo | Chat GLM4 Air |
|---|---|---|---|---|---|---|---|---|---|---|
| (0, 1] | 33.33 | 37.78 | 64.29 | 62.71 | 47.62 | 62.79 | 22.22 | 37.14 | 28.89 | 47.62 |
| (1, 5] | 45.95 | 50.40 | 55.50 | 60.08 | 44.08 | 53.99 | 37.24 | 40.55 | 37.70 | 48.06 |
| (5, 15] | 51.85 | 36.42 | 40.76 | 49.44 | 35.71 | 40.65 | 28.37 | 29.71 | 20.33 | 48.42 |
| (15, 30] | 46.67 | 25.00 | 27.59 | 43.14 | 22.22 | 44.64 | 8.11 | 38.89 | 17.14 | 48.89 |
| Overall | 46.59 | 41.97 | 47.90 | 55.18 | 49.89 | 40.71 | 30.74 | 36.71 | 30.55 | 48.28 |

bottom plot (filling the outputs of previous actions as parameters). The accuracy drop ranges from 2.55% (`GPT-3.5-turbo`) to 15.39% (`Deepseek-2-Chat`).

- **For existing less intelligent LLMs errors mainly stem from incorrect output formats and wrong API selections.** Figure 6b shows error types for tasks requiring outputs from previous actions. It can be seen that powerful LLMs like `Gemini-1.5-Pro` rarely make format errors, whereas the less intelligent models frequently make mistakes in output format and API selection.

The results from Recognition of Need for Input (Section 3.3.3) lead us to the following conclusions:

- **All agents perform poorly at recognizing necessary system and user inputs when required.** Overall, all agents have weak recognition capabilities, with accuracy ranging between 30.55% (`GPT-3.5-turbo`) and 55.18%(`Deepspeed-2-coder`). Larger LLMs such as `Deepspeed-2-chat` (236B) still demonstrate better recognition accuracy.

# 5 CONCLUSION

In this paper, we introduce SHORTCUTSBENCH, a benchmark for evaluating API-based agents. To the best of our knowledge, SHORTCUTSBENCH is the most realistic, rich, and comprehensive benchmark of its kind. Our findings indicate that for agents built on the most advanced LLMs, the primary bottleneck is API selection. For the most cost-effective LLMs, there is considerable room for improvement in both API selection and parameter filling. Additionally, we identified a significant deficiency in the agents' awareness of requesting necessary input.

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

## A  APPENDIX / SUPPLEMENTAL MATERIAL

### A.1  DATASET ACQUISITION

In this section, we introduce more details about the dataset acquisition introduced in Section 3.1.

Regarding data acquisition, we first use search engines to identify popular public shortcut-sharing sites (① in Figure 1). We found a total of 14 sites. These sites include:

Table 5: 14 shortcut-sharing sites, including names, URLs, and shortcut counts.

| # | Site Name | URL | Count |
|---|-----------|-----|-------|
| 1 | Matthewcassinelli | `https://matthewcassinelli.com` | 1535 |
| 2 | Routinehub | `https://routinehub.co` | 6860 |
| 3 | MacStories | `https://www.macstories.net/shortcuts` | 4993 |
| 4 | ShareShortcuts | `https://shareshortcuts.com` | 2395 |
| 5 | ShortcutsGallery | `https://shortcutsgallery.com` | 4269 |
| 6 | iSpazio | `https://shortcuts.ispazio.net` | 115 |
| 7 | Jiejingku | `https://jiejingku.net` | 3347 |
| 8 | SSPai | `https://shortcuts.sspai.com` | 145 |
| 9 | Jiejing.fun | `https://jiejing.fun` | 84 |
| 10 | Kejicut | `https://www.kejicut.com` | 37 |
| 11 | RCuts | `https://www.rcuts.com` | 133 |
| 12 | Sharecuts | `https://sharecuts.app` | 2395 |
| 13 | Siri-shortcuts | `https://www.siri-shortcuts.de` | 15 |
| 14 | Reddit | `https://www.reddit.com/r/shortcuts` | 100 |
| | | **Total (After Deduplication):** | 8675 |

We can obtain shortcuts from these sites. Specifically, each dataset includes the "shortcut name" (`NameInStore`), "function description" (`DescriptionInStore`), "shortcut type" (`CategoryInStore`), and most importantly, the "iCloud link" (Apple, 2024a). Additionally, it includes less important data such as the number of downloads (`Downloads`), favorites (`Favorites`), reads (`Reads`), and ratings (`Rates`). All shortcuts include `NameInStore` and `DescriptionInStore`, while the availability of other fields varies slightly depending on the specific shortcut-sharing site.

We then downloaded the shortcut source file by "iCloud link" and performed deduplication based on both iCloud links and the actual shortcut content (i.e., action sequences) to ensure the uniqueness of each shortcut in the final dataset (② in Figure 1). For details on downloading source files via iCloud links, please refer to our open-source code repository. We do deduplication because shortcuts sharing sites store shortcuts as iCloud links, which often results in the same shortcut appearing in multiple sharing-site. Additionally, shortcuts linked by these iCloud links could have identical content, making deduplication essential to ensure that each shortcut in the final dataset was unique.

We then extracted the app name using the field `WFWorkflowActionIdentifier` from the shortcut source file and downloaded the associated apps (③ in Figure 1). Shortcuts are composed of a series of shortcut API calls, referred to as Actions. An example of a typical shortcut is shown in Figure 7. Each shortcut API call is identified by a name, which usually includes the app's identifier, such as `com.openai.chat`, and the Intent name, such as `AskIntent`. For most API names, the segment before the last dot represents the app name, while the segment after denotes the Intent name. We semi-automatically extracted all app names to streamline the app download process.

We download these apps from various sources:

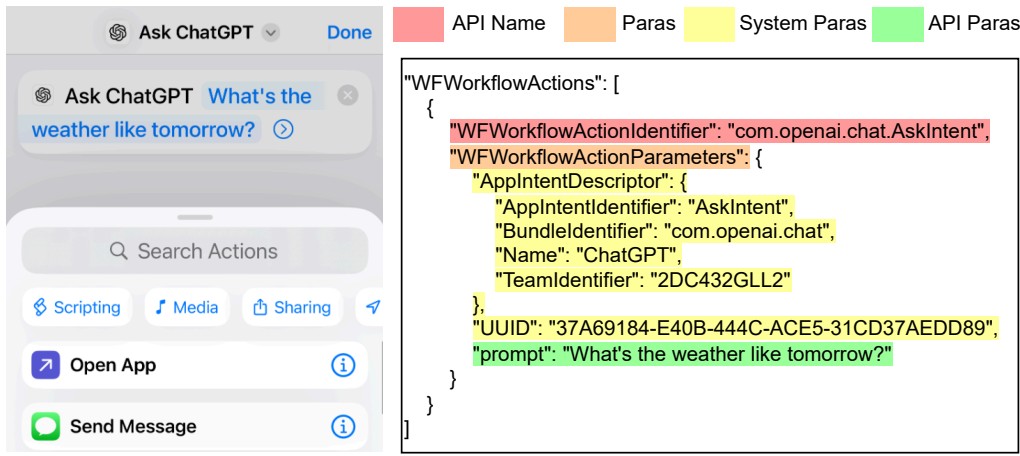

Figure 7: An example of a shortcut: `Ask ChatGPT`.

- Apps from the macOS or iOS App Store: We downloaded a variety of applications directly from Apple's official platforms. This provided us with a vast selection of apps that are widely used and trusted by users.

- System apps like *Keynote* from paths `/Applications/` and `/System/Application/` on macOS: These are pre-installed applications integral to the operating system. Including them ensured that our dataset covered essential tools commonly used by macOS users.

- Third-party apps from the official websites of the apps: To include software not available through the App Store, we downloaded apps from their official websites. This allowed us to capture a broader range of functionalities offered by third-party developers.

During the downloading process, we also excluded some legacy apps that are no longer maintained and 12 paid apps to avoid potential licensing issues and focus on applications readily accessible to the general public.

Then we managed to extract APIs from the downloaded apps (④ in Figure 1). The APIs are mainly from intent definition file `${filename}.actionsdata` from *AppIntent* (Apple-Inc., 2024b) framework and `${filename}.intentdefinition` from *SiriKit* (Apple-Inc., 2024c) framework. We extracted all APIs involved in the apps. During the extraction, we perform deduplication of APIs based on manually crafted rules as an app may have multiple duplicate API definition files with the same API definition.

We perform deduplication to streamline API definitions, minimize redundancy, and ensure compatibility across frameworks, addressing inconsistencies introduced by the coexistence of `SiriKit` and `AppIntents`. `SiriKit`, introduced in 2016 with iOS 10, enabled applications to integrate with Siri for voice command interactions. In 2022, Apple launched `AppIntents` with iOS 16, providing a more modern and flexible approach to defining and handling app intents. `AppIntents` facilitate integration with Siri, Shortcuts, widgets, and more. To encourage adoption, Apple has provided migration tools for developers transitioning from `SiriKit`. However, some apps still rely on the `SiriKit`. Under `SiriKit`, developers use `$filename.intentdefinition` files, while the `AppIntents` relies on `$filename.actionsdata` files. These files define APIs corresponding to actions in Shortcuts. Apps may include only `$filename.intentdefinition` files, only `$filename.actionsdata` files, or both, potentially leading to redundancy in API definitions. To address this, we have implemented a set of rules to reduce API definition files and ensure API uniqueness.

Additionally, for app *Shortcuts*, which are deeply integrated with Apple's operating system, we need to obtain their API definition files `WFActions.json` from system path `/System/Library/PrivateFrameworks/WorkflowKit.framework/` on macOS, instead of extracting it from the app itself.

Subsequently, we further filtered the shortcuts based on criteria such as whether the associated apps is paid app, whether the apps were outdated, and whether the APIs were deprecated. Additionally, we imported all shortcuts into the macOS Shortcuts app to ensure they were functional. These steps were repeated multiple times.

Finally, as shown in Table 2, we get 88 apps from various categories such as "Health & Fitness", "Developer Tools", and "Lifestyle". These apps in total include 1414 APIs, including all of 556 APIs (Not all APIs have been used in Shortcuts) involved in 7627 shortcuts.

The approximate time spent on each step of the process is outlined below:

- Shortcut site collection: Approximately 3 days, completed entirely manually.

- Link scraping using Selenium: Around 2 weeks, requiring custom scripts for each site.

- Shortcut deduplication, API validity checks, and shortcut functionality validation: Approximately 4 weeks. Deduplication: Automated using iCloud links and content cleaning.

- API validity checks: Performed manually. Shortcut validity checks: A mix of automated and semi-automated methods. Automated filtering was conducted using Apple Scripts to execute shortcuts for preliminary filtering, followed by manual validation through importing shortcuts into the Shortcuts app.

Additional manual and automated checks were conducted throughout the process but are not detailed here.

## A.2   DATASET CONSTRUCTION

The API definition files extracted from the app exist in two forms: the `${filename}.intentdefinition` files as indicated by the `Sirikit` framework and the `${filename}.actionsdata` files as indicated by the `App Intent` framework. Additionally, Apple's first-party apps provide a third type of definition file, `WFActions.json`. All three file formats provide "API description", "API name", "parameter names", "parameter types", "default value", "return value type", and "return value name", but differ in their file format. We give a sample from each of the three different file formats, as shown in Figure 8.

We construct queries based on existing action sequences and APIs. To ensure the quality of these queries, we utilize the natural language workflow descriptions unique to shortcuts. When generating queries, we require the model to naturally include primitive data type parameters and enum data types needed for API calls. This helps us evaluate the agent's ability to handle primitive parameters. We do not require the inclusion of complex data types in the queries, as they are difficult to convert to text and challenging to evaluate. To ensure high-quality query generation, we use the state-of-the-art LLM, `GPT-4o` (OpenAI, 2024). The prompt templates used for generating queries are provided in Figure 9.

To ensure the quality of shortcuts, we filtered them based on criteria such as whether the associated apps were paid, outdated, or relied on deprecated APIs. Additionally, all shortcuts were imported into the macOS Shortcuts app to verify functionality. Deduplication and error-checking processes were carried out throughout the entire data collection phase.

For ensuring the quality of generated queries, following prior work (Qin et al., 2024), we conducted a preliminary experiment with three LLMs: `GPT-4o`, `GPT-3.5`, and `Gemini-1.5-Pro`, on a dataset of 100 samples. Human evaluators rated `GPT-4o` as generating the highest-quality queries, outperforming the other two models. `GPT-4o` excelled in accurately identifying required parameters and providing clear query descriptions, meeting our criteria in 94 out of 100 cases. This superior performance can largely be attributed to the natural language workflow descriptions. While we acknowledge that not all queries may fully meet our requirements, we believe our approach is reasonable. Similar works, such as ToolLLM, rely on GPT for large-scale query and action sequence generation without guaranteeing complete accuracy.

---

**(1) com.openai.chat.AskIntent** (prompt: String, newChat: Boolean, model: ModelEntity, continuous: Boolean) -> Ask ChatGPT: String

**(2) Parameters** [parameter name (default value): title. parameterDescription]:
  **(2.1) prompt**: Message. Message to send to ChatGPT
  **(2.2) newChat** (0): Start new chat. Indicates whether a new chat should be started
  **(2.3) model** (default): Model. Model to use with the new chat
  **(2.4) continuous** (0): Continuous chat. Whether to enable back-and-forth chat or complete the Shortcut immediately after response

**(3) Return Value** [return value name: resultValueName. displayTypeName]:
  **(3.1) Ask ChatGPT**: None

**(4) Description** [title + description + actionSummary]:
  **(4.1) title**: Ask ChatGPT
  **(4.2) description**: This action will send a single message to a chat with ChatGPT and return the response.
  **(4.3) actionSummary**: Search for ${query}

---

**(1) is.workflow.actions.getrichtextfromhtml** (WFHTML: WFStringContentItem) -> Rich Text from HTML: public.html

**(2) Parameters** [parameter name (default value): DescriptionInput]:
  **(2.1) WFHTML**: HTML

**(3) Return Value** [return value name: DescriptionResult]:
  **(3.1) Rich Text from HTML**: None

**(4) Description** [Name + DescriptionSummary + ParameterSummary]:
  **(4.1) Name**: Make Rich Text from HTML.
  **(4.2) DescriptionSummary**: Takes the inputted HTML and turns it into rich text, which can then be converted to other formats.
  **(4.3) ParameterSummary**: Make rich text from ${WFHTML}

---

**(1) com.ulyssesapp.mac.ULInsertTextIntent** (sheet: SheetReference (Object), text: String, format: TextFormat (Enum), position: TextPosition (Enum)) -> Result: None

**(2) Parameters** [parameter name (default value): INIntentParameterDisplayName. INTypeDisplayName]:
  **(2.1) sheet**: Sheet. Sheet Reference
  **(2.2) text**: Content.
  **(2.3) format**: None
  **(2.4) TextPosition**: None

**(3) Return Value** [return value name: INIntentResponseParameterDisplayName]:
  **Result**: None

**(4) Description.** [INIntentTitle + INIntentDescription + INIntentParameterCombinationTitle]:
  **(4.1) INIntentTitle**: Add Text to Sheet.
  **(4.2) INIntentDescription**: Adds text to an existing sheet in Ulysses.
  **(4.3) INIntentParameterCombinationTitle**: Add ${text} to ${sheet}

---

Figure 8: We randomly selected three samples from three different definition files, as shown in the upper (`${filename}.actionsdata`), middle (`WFActions.json`), and lower (`${filename}.intentdefinition`) figures. The content in brackets represents different field names. In practice, there are various details to handle, such as name prefixes and missing fields. For complete details, please refer to our open-source code.

---

**SYSTEM_PROMPT_TEMPLATE:**

Shortcut consist of a sequence of actions, each is an API call, to execute user-provided queries.
As a user-friendly and patient inquirer, you need to craft a query based on the provided shortcut. This query, formatted as a question, should describe the task a user wants to complete and adhere to the following criteria:

1. The problem described in the query must be solvable using the shortcut.

2. The query should include all required parameters from the shortcut.

3. The query should be naturally phrased, integrating parameters seamlessly into the question rather than listing them separately.

For each shortcut command, I will provide you with five fields:

1. 'RecordName': The name of the shortcut, briefly describing its function.

2. 'Description of the Shortcut Workflow': A description of the entire action workflow of the shortcut.

3. 'Comments': Optional. Notes from the shortcut's developer, which may describe its functions or other features.

4. 'Description in Store': A description of the shortcut's functionality provided in the shortcut store.

5. 'API Description List': Detailed descriptions of the APIs involved in the shortcut.

You should rely primarily on the 'Description of the Shortcut Workflow' and 'API Description List', and refer to 'RecordName', 'Comments', and 'Description in Store' to formulate the final query.

**USER_PROMPT_TEMPLATE:**

Below are the five fields I provide to you:

1. 'RecordName': {RecordName}

2. 'Description of the Shortcut Workflow': {DescriptionoftheShortcutWorkflow}

3. 'Comments': {Comments}

4. 'Description in Store': {DescriptionInStore}

5. 'API Description List': {APIDescriptionList}

Please generate a query based on these details. Alongside the query, provide the shortcut's name and a description of its functionality using the following JSON format:

```
{
 "shortcut_name": "ThisIsShortcutName",
 "shortcut_description": "ThisIsShortcutDescription",
 "query": "ThisIsQuery"
}
```

Do not output any other content; your response should only be in this JSON format. Do not simply repeat the shortcut workflow. Parameters not surrounded by {{}} should not appear in the generated query. Output the JSON directly without using "'json XX"' to enclose it.
Note again, you should include all required parameters in the generated query. Please give your answer in English.

---

Figure 9: System and user prompt templates for query generation based on a shortcut

## A.3 TASK DEFINITION AND METRICS

Considering the context limitations of LLMs, we excluded shortcuts longer than 30 and parts using the API `is.workflow.actions.runworkflow` to call other shortcuts. While these shortcuts remain in our open-source dataset, they will not be included in the evaluation. We aim to study

the performance of agents on queries of varying difficulties. As shown in Table 3, we categorize SHORTCUTSBENCH into 4 difficulty levels and 8 task types based on $|aseq_i|$ and "shortcut type" (Section 3.1), respectively.

In calculating the length of shortcut commands, we do not simply count the number of actions within the shortcut. Instead, we apply a specialized approach. Initially, certain actions that do not contribute meaningful operations, such as `is.workflow.actions.comment` and `is.workflow.actions.alert`, which are akin to comments in programming, are excluded. Furthermore, we disregard the length of certain control flow statements, including `is.workflow.actions.conditional`, `is.workflow.actions.choosefrommenu`, `is.workflow.actions.repeat.count`, `is.workflow.actions.repeat.each`. For branching statements, we consider the length of the longest branch, rather than the cumulative length of all branches.

When categorizing shortcuts, we first analyzed all available categories from the `CategoryInStore` field in the collected data. We then classified the shortcuts into 8 categories, referencing with the classification of apps on the Apple App Store (app). The categories are as follows:

1. Productivity & Utilities

2. Health & Fitness

3. Entertainment & Media

4. Lifestyle & Social

5. Education & Reference

6. Business & Finance

7. Development & API

8. Home & Smart Devices

Subsequently, I employed a language model to categorize all shortcuts using the prompt shown in Figure 10.

### A.4    PERFORMANCE ABOUT API SELECTION

Following existing work (Huang et al., 2024b; Patil et al., 2024; Xu et al., 2024), we use the accuracy of API selection as the metric. The accuracy is calculated as the number of correct API selections $m_p$ divided by $n_p$. Specifically, each time we predict an action $b_j, 1 \leq j \leq |aseq_i|$, we provide the agent with all the correct historical actions $\{a_1, a_2, ..., a_{j-1}\}$. We then require the agent to predict the next action. All actions predicted by the agent form the prediction sequence $bseq_{p,i}$. This method is similar to the next token prediction (NTP) in LLMs, effectively preventing a cascade of errors in subsequent action predictions due to a single incorrect prediction. During the prediction, when encountering special actions such as branching and looping, we skip predicting these actions and directly add them to the historical actions.

Specifically, when calculating the precision of API selection, we do not consider the contributions of control statements such as branches and loops. This avoids the unreasonable requirement for the agent to invoke "branch APIs" or "loop APIs" in the next action. The agent should inherently possess the ability to correctly understand and act according to the conditions dictated by branches and loops. In addition to excluding the contributions of these control statements, we also disregard contributions from `is.workflow.actions.comment` and `is.workflow.actions.alert`, effectively removing these non-operative commands from the history of actions provided to the agent.

### A.5    EFFECTIVENESS OF API PARAMETER VALUE FILLING

To further ensure that the corresponding parameters are indeed included in the queries during evaluation, we used the LLM to filter these parameters further, ensuring their presence in the queries. Detailed prompts can be found in Figure 11.

**SYSTEM_PROMPT_TEMPLATE:**
Shortcut consist of a sequence of actions, each is an API call, to execute user-provided queries.
As a friendly and patient assistant, you need to categorize the provided shortcut into one of the following eight categories:

1. Productivity & Utilities

2. Health & Fitness

3. Entertainment & Media

4. Lifestyle & Social

5. Education & Reference

6. Business & Finance

7. Development & API

8. Home & Smart Devices

For each shortcut command, I will provide you with five fields:

1. 'RecordName': The name of the shortcut, briefly describing its function.

2. 'Description of the Shortcut Workflow': A description of the entire action workflow of the shortcut.

3. 'Comments': Optional. Notes from the shortcut's developer, which may describe its functions or other features.

4. 'Description in Store': A description of the shortcut's functionality provided in the shortcut store.

5. 'API Description List': Detailed descriptions of the APIs involved in the shortcut.

You should rely primarily on the 'Description of the Shortcut Workflow' and 'API Description List', and refer to 'RecordName', 'Comments', and 'Description in Store' to give the final category.

**USER_PROMPT_TEMPLATE:**
Below are the five fields I provide to you:

1. 'RecordName': {RecordName}

2. 'Description of the Shortcut Workflow': {DescriptionoftheShortcutWorkflow}

3. 'Comments': {Comments}

4. 'Description in Store': {DescriptionInStore}

5. 'API Description List': {APIDescriptionList}

Please give the category on these details. Alongside the category, provide the shortcut's name and a description of its functionality in English using the following JSON format:

```
{
    "category": "category",
    "english_name": "ThisIsShortcutName",
    "english_functionality": "ThisIsFunctionality"
}
```
Do not output any other content; your response should only be in this JSON format.

Output the JSON directly without using "'json XX"' to enclose it. Please give your answer in English.

Figure 10: System and user prompt templates for categorizing shortcuts based on their functionalities

---

**SYSTEM_PROMPT_TEMPLATE:**
Your task is to classify the parameters I provide based on user queries, API information, and API calls (also known as actions).

User query describes the task the user wants to accomplish.

Information about the API definition includes the API name, parameter names, parameter types, default values, return value names, and return value types. Parameters are identified by 'Parameters' and explained. The return value names and return value types are identified by 'Return Values'. The API's brief and detailed descriptions are marked by 'Description'. The natural language description of the API is marked by 'ParameterSummary'.

Completing the user query requires a series of API calls, each API call needs the correct and appropriate parameters. We have pre-selected possible parameters that may appear in the query.

Please note, you must classify these pre-selected parameters based on the user query. Each parameter can generally be classified into the following categories:

1. Precise parameter: Parameters stated by users in the query, or those implicitly indicated in the query but can be accurately inferred by combining the query and the API definition.

2. Not precise parameter: Parameters not stated by users in the query and cannot be accurately inferred even with the combination of the query and the API definition.

Note! Note! Note! all precise parameters must be clearly or implicitly specified in the query.

**USER_PROMPT_TEMPLATE:**
The user query is: {query}
Information about the API definition is provided below: {api_desc}
The API call is: {API_call} The pre-selected possible parameters that may appear in the query are listed below: {possible_paras}

Output the classification in the following format:

```
{
    para_name1: {
        para_name1: para_type1,
        "reason1": The reason
    },
    para_name2: {
        para_name2: para_type2,
        "reason2": The reason
    },
    ...
}
```

Do not output any additional content; only output a JSON. Do not enclose your output with "'json XXX"'.
Note! Note! Note! all precise parameters must be clearly or implicitly specified in the query.

Figure 11: System and user prompt templates for classifying parameters based on user queries and API definitions

## A.6 RECOGNITION OF NEED FOR INPUT

In the shortcut, a parameter can be set to `ExtensionInput`, indicating that the parameter requires a file provided by the user, or `CurrentDate`, indicating that the parameter needs to retrieve the date from the system. Similarly, `Clipboard` indicates that the parameter should obtain content from the clipboard, and `DeviceDetails` implies that the parameter needs to access certain information about the user's device. Lastly, `Ask` denotes that the parameter requires user authorization or

essential input from the user. A typical example is shown in Figure 12, where the action uses the `is.workflow.actions.getmyworkflows` API. The `Folder` parameter is set to `Ask`, indicating that this parameter requires input provided by the user.

```
{
"WFWorkflowActionIdentifier": "is.workflow.actions.getmyworkflows",
"WFWorkflowActionParameters": {
  "Folder": {
    "Value": {
      "Type": "Ask"
    },
    "WFSerializationType": "WFTextTokenAttachment"
  },
  "UUID": "E5F695A5-9DD3-4720-84D2-9AB0AD457908"
}
}
```

Figure 12: An example of `Ask` parameter.

## A.7 SETUP

Following existing work (Huang et al., 2024b; Qin et al., 2024; Li et al., 2023), we slightly modified the ReACT (Yao et al., 2023) templates to construct the API-based agents. The templates used in our experiments are as shown in Figure 13.

## A.8 RESULT ANALYSIS

Table 6: Pricing, Testing Instances, and Actual Costs of Popular AI Models. (07-22-24). Except for `gemini-1.5-pro`, which was randomly tested on 800 instances due to cost considerations, all other LLMs were tested across all datasets. However, the number of successful tests varied slightly due to factors such as context length, safety reviews, and etc. The cost of testing primarily stems from inputs, as we continuously feed historical actions into the LLM for evaluation, and all historical conversations are billed repeatedly (OpenAI Community, 2023).

| Model Name | Price / 1M tokens | Instances | Estimate Cost ($) |
|---|---|---|---|
| gemini-1.5-pro | $3.50 / $10.50 | 801 | 592 |
| gemini-1.5-flash | $0.35 / $1.05 | 5295 | 391 |
| qwen2-72b-instruct | $0.70 / $1.40 | 5216 | 800 |
| qwen2-57b-a14b-instruct | $0.49 / $0.98 | 5368 | 580 |
| GPT-4o-mini | $0.15 / $0.60 | 5320 | 100 |
| gpt-3.5-turbo | $0.50 / $1.50 | 5463 | 500 |
| deepseek-chat | $0.14 / $0.28 | 5319 | 90 |
| deepseek-coder | $0.14 / $0.28 | 5317 | 90 |
| GLM-4-Air | $0.14 / $0.14 | 5330 | 110 |
| **Total Cost** | | | **3253** |

Among them, `gemini-1.5-pro` (tested with 801 instances) and `gemini-1.5-flash` (tested with 5,295 instances) incurred a total cost of $801, with `gemini-1.5-flash` accounting for approximately $391 and `gemini-1.5-pro` approximately $592. The costs for `qwen2-72b-instruct` (tested with 5,216 instances) were about $800, `qwen2-57b-a14b-instruct` (tested with 5,368 instances) around $580, and `GPT-4o-mini` (tested with 5,320 instances) approximately $50. `gpt-3.5-turbo` (tested with 5,463 instances) cost approximately $500. The combined expenses for `deepseek-chat` (tested with 5,319 instances) and `deepseek-coder` (tested with 5,317 instances) were roughly $180, while `GLM-4-Air` cost about $110.

---

**SYSTEM_PROMPT_TEMPLATE:**
You are AutoGPT. Your task is to complete the user's query using all available APIs.

First, the user provides the query, and your task begins.
At each step, you need to provide your thought process to analyze the current status and determine the next action, with an API call to execute the step. After the call, you will receive the result, and you will be in a new state. Then, you will analyze your current status, decide the next step, and continue...
After multiple (Thought-Call) pairs, you will eventually complete the task.

Below are all the available APIs, including the API name, parameter names, parameter types, default values, return value names, and return value types.
{all_api_descs}

For each step, use only one API. Strictly follow the JSON format below for your output and do not include any irrelevant characters.

```
{
 "Thought": "Your analysis of what to do next",
 "WFWorkflowActionIdentifier": "The API name you call",
 "WFWorkflowActionParameters": {
     "parameter name": "parameter value"
 }
}
```

WFWorkflowActionParameters are the parameters required for the API call. The parameter value might be:

1. basic data types like string, integer, float, or boolean.

2. output from previous API call.

3. input from the system or the user, including file provided by the user.

4. Previously defined variable names.

5. If the parameter is of type string, you can also combine the output of a previous action, input from the system or the user, with a string.

6. If the output of a previous action is an Object type, or if you need to use input from the system or the user, you can utilize specific properties from the previous action's output.

**USER_PROMPT_TEMPLATE:**
The user query is: {query}
The history actions and observations are as follows: {history_actions}

Please continue with the next actions based on the previous history. Do not output any other content; your response should only be in this JSON format.
You should only output one action at a time.

Figure 13: System and user prompt templates for executing API calls based on user queries

The cost analysis indicates a notable range in efficiency and value for money. Models like `deepseek-chat` and `deepseek-coder` show excellent cost-effectiveness, particularly suitable for high-volume, low-cost deployments. In contrast, models like `gemini-1.5-pro` and `gemini-1.5-flash` reflect higher costs, but they offer superior performance.

## A.9 ADDITIONAL RESULTS ON EVALUATION ON SPECIALIZED API-CALLING FINE-TUNED LLMS

To comprehensively evaluate the effectiveness of fine-tuning LLMs for agent-based scenarios, we extended our comparison to include specialized AI agent models such as `AgentLM` (Zeng et al., 2024) and `xLAM` (Zhang et al., 2024), alongside general-purpose LLMs like `Qwen-2-72B` Qwen

Table 7: Performance Comparison of Fine-Tuned and Base LLMs

| Model | LLaMA-2-7B AgentLM-7B | | LLaMA-2-13B AgentLM-13B | | LLaMA-2-70B AgentLM-70B | |
|---|---|---|---|---|---|---|
| | Base | Fine-Tuned | Base | Fine-Tuned | Base | Fine-Tuned |
| Context Length | 4k | 4k | 4k | 4k | 4k | 4k |
| API Selection | **32.75** | 24.95 | **38.59** | 30.57 | **52.13** | 43.41 |
| | 12.57 | **17.43** | 25.99 | **35.48** | **43.08** | 37.34 |
| | 23.70 | **32.78** | 35.45 | **40.78** | **57.76** | 39.22 |
| API Parameter Value Filling | **51.11** | 34.88 | 30.77 | **31.82** | **72.86** | 52.83 |
| | 32.14 | **35.71** | 12.50 | **40.98** | **45.05** | 37.31 |
| | 0.00 | 0.00 | 11.11 | **40.00** | 6.67 | **11.11** |
| Recognition of Need for Input | 7.41 | **10.53** | 5.00 | **5.26** | **21.74** | 0.00 |
| | 12.50 | 4.17 | 0.00 | **3.57** | 7.69 | **9.09** |
| | 0.00 | 0.00 | 0.00 | 0.00 | 0.00 | 0.00 |
| Model | Mistral-7B xLAM-7b-r | | Mixtral-8x7b xLAM-8x7b-r | | LLaMA-2-70B Lemur-70B-Chat-V1 | |
| | Base | Fine-Tuned | Base | Fine-Tuned | Base | Fine-Tuned |
| Context Length | 32k | 32k | 32k | 32k | 4k | 4k |
| API Selection | 17.03 | **70.56** | 70.21 | **85.29** | 52.13 | **52.45** |
| | 3.27 | **50.82** | 49.15 | **51.72** | 43.08 | **47.08** |
| | 0.50 | **66.33** | 29.92 | **29.67** | 57.76 | **60.39** |
| API Parameter Value Filling | 40.00 | **66.67** | 53.45 | **68.93** | **72.86** | 45.45 |
| | 40.54 | **59.32** | 33.72 | **46.76** | **45.05** | 36.49 |
| | 11.11 | **31.82** | 27.20 | **42.71** | 6.67 | **26.92** |
| Recognition of Need for Input | 0.00 | **5.56** | 6.25 | **11.90** | **21.74** | 16.00 |
| | 0.00 | **2.56** | 19.27 | **19.91** | **7.69** | 3.57 |
| | 0.00 | 0.00 | 0.00 | 0.00 | 0.00 | 0.00 |

(2024) and `LLaMA-3-70B` Meta (2024). This was aimed at understanding the applicability of fine-tuning strategies for API-based agents.

We conducted experiments with additional 11 LLMs, comprising 6 models fine-tuned specifically for agent scenarios and 5 base models. The fine-tuned models include the `AgentLM` series (AgentLM-7B, AgentLM-13B, AgentLM-70B), the `xLAM` series (xLAM-7b-r, xLAM-8x7b-r), and `Lemur` (Lemur-70B-Chat-V1). The base models include various configurations of `LLaMA-2` and `Mistral` from Hugging Face repositories.

The fine-tuning processes of these LLMs varied: the `AgentLM` series was fine-tuned on 1866 traces of simple code data, `Lemur` utilized scripting and interpreted languages from The Stack dataset, and the xLAM series was specialized for API-based agents with a predefined format for API calls. To ensure standardized evaluation, outputs were processed using `gpt-4o-mini`, with manual verification confirming the accuracy of conversions. Due to context length limitations (4k tokens for most models and 32k tokens for `xLAM` series), results are reported for the first 3 difficulty levels.

Experimental results are summarized in Table 7.

A.9.1    RECOGNITION OF NEED FOR INPUT

All evaluated LLMs performed exceptionally poorly in the recognition of need for input, significantly lagging behind the general LLMs mentioned in the ShortcutsBench. Compared to the performance of the base LLMs before fine-tuning, neither the AgentLM and Lemur series models fine-tuned on code nor the xLAM series models fine-tuned specifically for API-based agents showed any improvement

in this aspect. This suggests the need for targeted fine-tuning specifically aimed at enhancing this capability.

### A.9.2 API PARAMETER VALUE FILLING

The evaluated LLMs perform significantly worse in API Parameter Value Filling compared to the general LLMs mentioned in ShortcutsBench. The general LLMs achieve accuracies of 94.29%, 83.92%, and 75.53% across three difficulty levels, whereas the best results from the evaluated models only reach accuracies of 72.86%, 59.32%, and 42.71%.

Compared to the base LLMs before fine-tuning, the code-based agents, including the AgentLM and Lemur models fine-tuned on code, did not achieve a general improvement in parameter-filling accuracy for API-based tasks and even exhibited performance degradation. In contrast, the API-based agents, such as the xLAM series fine-tuned specifically for API-based agents, showed a consistent improvement in parameter-filling accuracy. The existing fine-tuning methods for API-based agents (xLAM-7b-r / xLAM-8x7b-r) are effective for parameter filling in short sequences, but accuracy drops sharply as the context length increases.

### A.9.3 API SELECTION

Methods fine-tuned specifically for API-based agents, such as the xLAM series models, can significantly enhance agents' API selection accuracy. The xLAM series achieves reasonable accuracy in API selection, even compared to general LLMs. However, compared to xLAM-7b-r, the API selection accuracy of xLAM-8x7b-r is significantly lower, suggesting that existing fine-tuning methods specialized for agents may only achieve notable improvements on models with relatively weaker foundational capabilities.

Additionally, methods fine-tuned on code can still enhance agents' API selection accuracy, as demonstrated by the AgentLM-7b/13b and Lemur-70b-chat-v1 models. The performance drop of AgentLM-70b may be attributed to the fine-tuning dataset used for the AgentLM series lacking realism and complexity.

