# OpenReview forum: "ShortcutsBench: A Large-Scale Real-world Benchmark for API-based Agents"
_ICLR.cc/2025/Conference — ICLR 2025 Poster_

### Official Review · Reviewer_vqjw · 2024-10-30

**Soundness:** 2
**Presentation:** 2
**Contribution:** 3
**Rating:** 6
**Confidence:** 3

**Summary:**

The paper introduces ShortcutsBench, a benchmark to assess LLMs’ ability to call external APIs and create structured text (ex: JSON). The dataset was collected from the Shortcuts tool available on Apple devices. The authors perform the analysis of performance of a number of open and closed-source LLMs against ShortcutsBench.

**Strengths:**

[1] Analysis results in 4.2 are insightful and confirm the ranking of closed- and open-source LLMs on traditional general knowledge and reasoning benchmarks like MMLU.

[2] The artifacts are provided and well-organized.

**Weaknesses:**

[1] It is a technical report that lacks a scientific component. It should go to a software engineering conference, for example, International Conference on Software Engineering (ICSE) or IEEE/ACM International Conference on Automated Software Engineering (ASE). The paper can also go to Datasets and Benchmarks tutorials at AI conferences. Whereas the work is technically impressive and valuable for the software engineering community, there is no scientific value in the assessment of how good LLMs are able to use APIs. The problem at hand is purely technical and as a clear indicator for this is that over two thirds of references are links to web pages rather than scientific studies.

[2] The analysis of the performance of GPT 4o with the structured output capability was not performed.
https://openai.com/index/introducing-structured-outputs-in-the-api/

[3] The used language appeals to emotions rather than states the scientific value (ex: “We made great efforts to evaluate…”).

[4] Several references are outdated, pointing to arxiv whereas the paper has been already published at a conference (ex: ToolLLM).

**Questions:**

I suggest to submit to a more relevant conference, rather than to ICLR, as mentioned above.

---

> ### Author Response · Authors · 2024-11-14
>
> 1. It is a technical report that lacks a scientific component. It should go to a software engineering conference, for example, International Conference on Software Engineering (ICSE) or IEEE/ACM International Conference on Automated Software Engineering (ASE). The paper can also go to Datasets and Benchmarks tutorials at AI conferences. Whereas the work is technically impressive and valuable for the software engineering community, there is no scientific value in the assessment of how good LLMs are able to use APIs. The problem at hand is purely technical and as a clear indicator for this is that over two thirds of references are links to web pages rather than scientific studies.
>
> There are many excellent benchmark studies presented at top conferences such as ICLR, NeurIPS, and ACL. We are unsure what specific concerns you have about my work, and we are unclear whether your comments are directed at AI conferences or top software engineering conferences. If you have any concerns, please provide specific feedback, as it will help us address them more effectively.
>
> Regarding your comments, such as "It is a technical report that lacks a scientific component" and "there is no scientific value in assessing how well LLMs can use APIs", we must respectfully disagree. Our work establishes a pipeline for rapidly collecting real-world action sequences and provides findings that we believe will be helpful to others, backed by substantial analysis, thought, and effort.
>
> The top-tier conference papers dedicated entirely to datasets/benchmarks include, but are not limited to, the following:
>
> - (ICLR’24) SmartPlay : A Benchmark for LLMs as Intelligent Agents
> - (ICLR’24) AgentBench: Evaluating LLMs as Agents
> - (ICLR’24) ToolLLM: Facilitating Large Language Models to Master 16000+ Real-world APIs
> - (ICLR’24) MetaTool Benchmark for Large Language Models: Deciding Whether to Use Tools and Which to Use
> - (NeurIPS’24) FinBen: An Holistic Financial Benchmark for Large Language Models
> - (NeurIPS’24) ZSC-Eval: An Evaluation Toolkit and Benchmark for Multi-agent Zero-shot Coordination
>
> Evaluation has always been a critical component in guiding the development of AI, so we do not understand why you believe "there is no scientific value in assessing how well LLMs can use APIs". We would appreciate further clarification on what you define as "science" and "engineering", and we look forward to any insights you might offer.
>
> As for your comment, "The paper can also go to Datasets and Benchmarks tutorials at AI conferences", we would like to clarify that when we submitted to ICLR, we chose the Primary Area of "Datasets and Benchmarks", and the keywords we selected included "Benchmark". ICLR has accepted many high-quality benchmark evaluation papers.
>
> Regarding your concern that "two thirds of references are links to web pages", we must emphasize that the web pages we referenced were considered necessary to clearly explain our methodology. I apologize if this caused any inconvenience. We will strive to reduce the use of web citations and ensure that we present our work as clearly as possible with minimal references.
>
> Finally, thank you for your kind words about our paper being "technically impressive and valuable".

---

> ### Author Response · Authors · 2024-11-14
>
> 2. The analysis of the performance of GPT 4o with the structured output capability was not performed. https://openai.com/index/introducing-structured-outputs-in-the-api/
>
> Structured output refers to a function that controls the output format to ensure it is correctly formatted, such as in JSON. This is not directly related to the main focus of our research (API selection, parameter filling, and awareness in requesting necessary input). Furthermore, as mentioned in the paper, we did not conduct evaluations with GPT-4o due to budget constraints.

---

> ### Author Response · Authors · 2024-11-14
>
> 3. The used language appeals to emotions rather than states the scientific value (ex: “We made great efforts to evaluate…”).
>
> We apologize for any confusion caused by this statement. We will revise it in the next version.

---

> ### Author Response · Authors · 2024-11-14
>
> 4. Several references are outdated, pointing to arxiv whereas the paper has been already published at a conference (ex: ToolLLM).
>
> We apologize for not updating the correct citations in a timely manner. We will carefully review and ensure the accuracy of all citations in the next version.

---

> ### Author Response · Authors · 2024-11-14
>
> Thank you once again for your feedback. I truly hope that my responses above have helped clarify any concerns you may have. We would be very grateful if you could carefully consider your final rating. Please don’t hesitate to reach out with any additional questions or comments—I would be more than happy to assist further.

---

> ### Author Response · Authors · 2024-11-19
>
> We sincerely hope our response has addressed your concerns. We kindly ask you to carefully consider your scoring. If you have any further questions, we are always happy to assist.

---

> > ### Comment · Reviewer_vqjw · 2024-11-22
> >
> > In light of the answers by the authors, I have updated my score to "marginally below the acceptance threshold," meaning that if there is a consensus among other reviewers to accept the paper, I won't oppose it.

---

> > > ### Author Response · Authors · 2024-11-22
> > >
> > > Thank you for your understanding!

---

> ### Author Response · Authors · 2024-11-24
>
> Thank you for your understanding! We have thoroughly reviewed your comments and addressed them in the updated [PDF](https://openreview.net/pdf?id=kKILfPkhSz).
>
> ### Concern 3:
>
> We have removed the relevant statements from the text, particularly in the later paragraphs of the Introduction. Please take a look!
>
> ### Concern 4:
>
> We have removed most URL references in the text and used descriptive explanations instead. Additionally, we updated citations for all papers with formal references. For conferences like ICLR/NeurIPS, where no official references were available, we used the OpenReview versions of the papers.
>
> ### Additional Notes:
>
> As the discussion phase concludes on November 26 (AOE), we have implemented all the changes we committed to so far. However, there may still be some oversights, and we will use the remaining time to carefully review and refine our revisions.
>
> Thank you for your kind words about the technical strength of our work. We hope that after addressing your concerns, you will further consider your final score. If you have any additional questions, we are always happy to assist!

---

> ### Author Response · Authors · 2024-11-29
>
> We have conducted additional experiments on language models fine-tuned specifically for Agents, hoping to provide further insights. If you’re interested, feel free to check out the replies in XY2i’s thread.

---

> > ### Comment · Reviewer_vqjw · 2024-12-03
> >
> > I have updated my judgment to the acceptance level.

---

> > > ### Author Response · Authors · 2024-12-03
> > >
> > > Thank you for your understanding and support! Wishing you all the best!

---

### Official Review · Reviewer_2zSS · 2024-10-30

**Soundness:** 2
**Presentation:** 2
**Contribution:** 2
**Rating:** 6
**Confidence:** 3

**Summary:**

This paper introduces ShortcutsBench, which mines real-world APIs/action sequences from iOS shortcut app. They then synthesize tasks with language model, and test a range of model's ability on it. Results show the task remains challenging to today's LMs.

**Strengths:**

- Interesting approach: mining APIs/existing action sequence from Shortcut Apps makes a lot of sense, which is a resource previous works haven't tapped into.
- Comprehensive evaluation: the authors evaluated a wide range of LMs across Open and close source models.

**Weaknesses:**

1. Limited evaluation: The paper primarily assesses the model’s ability to choose correct actions based on ground-truth sequences but doesn’t evaluate its end-to-end task success rate (as done in, for example, AppWorld [1]). Experiments linking these aspects are missing.
2. No human validation: Given the synthetic nature of the benchmark, it’s uncertain whether all tasks are truly solvable or what the benchmark’s upper bound is. Including human performance as a reference would add clarity.

[1] AppWorld: A Controllable World of Apps and People for Benchmarking Interactive Coding Agents

**Questions:**

1. A notable feature of this benchmark seems to be scalability -- extracting real-world APIs and action sequences from Shortcut apps and sharing sites seems relatively easy. Providing quantitative details on dataset construction—such as time taken for each step and the actual extent of manual effort involved—would be nice.

---

> ### Author Response · Authors · 2024-11-14
>
> 1. *Limited evaluation: The paper primarily assesses the model’s ability to choose correct actions based on ground-truth sequences but doesn’t evaluate its end-to-end task success rate (as done in, for example, AppWorld [1]). Experiments linking these aspects are missing.*
>
> Thank you for your comment. We will consider building a relevant dynamic runtime environment on the macOS platform in future work. We have already undertaken some preliminary tasks, such as validating shortcut functionality, enabling apps exclusive to iOS to run on macOS via decryption, and translating API calls generated by LLMs into shortcut commands. To fully evaluate whether a shortcut is "successful", Apple Scripts are required to run the shortcuts and capture signals from the associated apps to determine the final status. Otherwise, manual evaluating would be necessary, and we regret that we cannot "manually verify" all model-generated shortcuts in the short term.
>
> Nevertheless,  we believe that ShortcutsBench offers distinct features and advantages compared to existing datasets/benchmarks, we believe we have provided a substantial amount of evaluation results, covering areas such as API selection, parameter filling, and feedback awareness in agent-based API execution. We hope these contributions will inspire future research in the field.
>
> We will also add a citation for AppWorld and discuss relevant similarities and differences in our paper.
>
> Thank you again for your valuable comment.

---

> ### Author Response · Authors · 2024-11-14
>
> 2. *No human validation: Given the synthetic nature of the benchmark, it’s uncertain whether all tasks are truly solvable or what the benchmark’s upper bound is. Including human performance as a reference would add clarity.*
>
> Regarding "*whether all tasks are truly solvable*", please refer to our response to reviewer "**XY2i**" on their first question. In summary, we implemented three steps to ensure the quality of our dataset, with both Step 2 and Step 3 involving manual verification. Since these shortcuts are available in the Shortcut Store specifically to address "real user needs" and have undergone human review, we believe there are no inherently "unsolvable" shortcuts.
>
> As for "*what the benchmark’s upper bound is*", we have not conducted a manual evaluation of API selection and parameter filling for each shortcut. However, we would like to note that none of the existing works referenced in Table 2, and also, AppWorld, as you mentioned, have performed this level of manual validation.
>
> We are very open to performing a manual validation of API selection and parameter filling. However, it is challenging to complete this in the short term. API calls require skilled individuals with specific knowledge of Shortcut programming, and parameter filling entails complex annotation tasks. Moreover, ShortcutsBench includes a substantial number of shortcuts, further adding to the workload.
>
> We sincerely appreciate your valuable feedback, and if you have any further questions, we are always happy to assist.

---

> ### Author Response · Authors · 2024-11-14
>
> 3. *A notable feature of this benchmark seems to be scalability -- extracting real-world APIs and action sequences from Shortcut apps and sharing sites seems relatively easy. Providing quantitative details on dataset construction—such as time taken for each step and the actual extent of manual effort involved—would be nice.*
>
> Thank you for your valuable feedback. Scalability is indeed a key feature of our framework. In fact, it’s entirely possible to set up a pipeline where shortcut programming is performed manually to generate action sequences, allowing us to efficiently construct a large number of action sequences.
>
> As for the specifics regarding the time required to build the dataset, we regret that we did not keep detailed records, but we can provide a general estimate:
> - **Collecting shortcut sites:** 3 days; this phase was entirely manual.
> - **Scraping links via Selenium automated testing tool:** 2 weeks; this step required custom code for each site.
> - **Shortcut deduplication, API validity check, and shortcut validity check:** 4 weeks; shortcut deduplication was automated through iCloud links and action content cleaning. API validity checks require manual verification. Shortcut validity checks combined with automated and semi-automated methods: automated filtering was conducted using Apple Scripts to run shortcuts for initial filtering, followed by manual loading of shortcuts into the Shortcuts app for verification.
>
> There were also various additional manual and automated checks, which we will not detail here.

---

> ### Author Response · Authors · 2024-11-14
>
> Thank you again for your valuable feedback. If you have any further questions, we are always here to help!

---

> ### Author Response · Authors · 2024-11-19
>
> Thank you for recognizing our work. I hope my previous responses have addressed your questions. If you have any new questions or if anything remains unclear, we’re always happy to provide further clarification.

---

> > ### Comment · Reviewer_2zSS · 2024-11-22
> >
> > Thank you for addressing my earlier comments.
> > On W2, I would still strongly recommend incorporating human baselines into this paper. Given the data construction process, there is no guarantee that the provided instructions contain sufficient information for even a highly capable system to achieve 100% accuracy. Without human baselines, it's unclear whether the current systems are genuinely underperforming or if there are inherent issues in the benchmark itself.
> >
> > That said, the benchmark is great for its realism and scalability, and I maintain my positive score of 6.

---

> > > ### Author Response · Authors · 2024-11-23
> > >
> > > Thank you for your recognition. We currently use a sampling-based manual verification approach, similar to ToolLLM, to ensure accuracy. To further guarantee “100%” accuracy on tested shortcuts, we will carefully consider your suggestion and assign professionals to determine the performance ceiling.

---

> ### Author Response · Authors · 2024-11-24
>
> Thank you for your timely response and encouraging feedback! We have thoroughly reviewed your comments and addressed them in the updated [PDF](https://openreview.net/pdf?id=kKILfPkhSz).
>
> ### Concern 1:
>
> We added a reference to AppWorld with a brief introduction in the second paragraph of Section 2: Related Work.
>
> ### Concern 2:
>
> In Section 3.2 (Query and action sequence), we elaborated on our efforts to ensure query quality, with further details available in Appendices A.1 and A.2:
>
> - The first paragraph discusses measures taken to maintain shortcut usability.
> - The final paragraph highlights actions to improve query accuracy.
>
> Additionally, the full processing logs and records are publicly accessible at https://anonymous.4open.science/r/ShortcutsBench.
>
> ### Concern 3:
>
> In Contribution 2 of the Introduction, we added the statement: “We hope this approach to dataset construction will inspire more researchers”. We added descriptions of the estimated time for each operation in Appendix A.1.
>
> ### Additional Notes:
>
> As the discussion phase concludes on November 26 (AOE), we have implemented all the changes we committed to so far. However, there may still be some oversights, and we will use the remaining time to carefully review and refine our revisions.
>
> Thank you once again for your positive feedback! If you have any further questions, I’d be happy to assist.

---

> ### Author Response · Authors · 2024-11-29
>
> We have conducted additional experiments on language models fine-tuned specifically for Agents, hoping to provide further insights. If you’re interested, feel free to check out the replies in XY2i’s thread.

---

### Official Review · Reviewer_AydN · 2024-11-02

**Soundness:** 3
**Presentation:** 2
**Contribution:** 3
**Rating:** 6
**Confidence:** 4

**Summary:**

This paper introduces a new benchmark ShortcutsBench, which evaluates agents' capabilities in solving tasks through API calling. They compare multiple API-based benchmarks, and showcase why ShortcutsBench contributes to the assessing agents' API calling abilities. The paper evaluates several API-based agents constructed based on the ReACT framework. The authors perform detailed analysis on the evaluation results.

**Strengths:**

- The authors introduce ShortcutsBench, which is a more holistic benchmark that contains real APIs, well-designed queries and actions. This could contribute to better evaluation of current agents's API calling capabilities in solving real-world tasks.
- The authors provide example instances from ShortcutsBench in the appendix, which helps understanding the types of tasks in this benchmark.
- The authors provide detailed analysis based on the evaluation results of several API-based agents.

**Weaknesses:**

- Section 3 is not well-elaborated. readers will benefit from clearer description of this process. For example, for (2), the authors say 'after duplicating based on icloud link, ....', it is not very clear what is duplicated and why this step helps. It would be good if the authors could refine their descriptions on their methodology.
- The authors cite each work too many times in the paper, for example, a research paper is is cited five times in one paragraph in Section 2. Referencing previous works is good practice, but referencing too many times affects readability. It would be good if they could remove repetitive references.
- The paper does not make it very clear what is their most important finding in the abstract/intro/conclusion. It would be good if they could highlight their most important findings in the abstract/intro/conclusion. For example, they could discuss how open source models perform comparably to closed source models on simpler tasks but not harder tasks.

**Questions:**

- Why is all API-based agents the authors evaluate based on the ReACT framework, it would be good if they could provide additional evaluation/analysis on other frameworks, such as CodeAct.

---

> ### Author Response · Authors · 2024-11-14
>
> 1. *Section 3 is not well-elaborated. readers will benefit from clearer description of this process. For example, for (2), the authors say 'after duplicating based on icloud link, ....', it is not very clear what is duplicated and why this step helps. It would be good if the authors could refine their descriptions on their methodology.*
>
> With the sentence "After deduplicating based on 'iCloud link' (Apple, 2024b), we got the source files of all 8675 shortcuts", we intend to convey the following:
>
> - During the data construction process, we extracted shortcuts from multiple different Shortcut repositories. Since these repositories store shortcuts as iCloud links, the same shortcut might appear in multiple repositories. Additionally, the actual shortcut content (i.e., the action sequences) linked by these iCloud links could also be identical. Therefore, we applied deduplication to ensure that each shortcut in the final dataset was unique.
>
> Beyond deduplication, we performed several additional operations, such as filtering out shortcuts dependent on non-existent APIs and importing all shortcuts into the Shortcuts app to verify their validity and functionality. For conciseness, only key steps are summarized in the paper. The complete process is available at https://anonymous.4open.science/r/ShortcutsBench, and further supplementary details can be found in the Appendix.
>
> If you have any further questions, please don’t hesitate to let us know. We would be glad to assist in any way we can.

---

> ### Author Response · Authors · 2024-11-14
>
> 2. *The authors cite each work too many times in the paper, for example, a research paper is is cited five times in one paragraph in Section 2. Referencing previous works is good practice, but referencing too many times affects readability. It would be good if they could remove repetitive references.*
>
> We apologize for any inconvenience this may have caused. Each cited paper is referenced multiple times because each has a distinct focus and contributes uniquely to different aspects of our discussion. In cases where methods fall under various classification schemes, multiple references to the same work allow for more accurate contextual alignment with each specific point.
>
> We apologize for any inconvenience caused by the extensive citations or repeated references across sections. For the final published version, we will adjust by moving some citations to the appendix to streamline the reading experience.

---

> ### Author Response · Authors · 2024-11-14
>
> 3. *The paper does not make it very clear what is their most important finding in the abstract/intro/conclusion. It would be good if they could highlight their most important findings in the abstract/intro/conclusion. For example, they could discuss how open source models perform comparably to closed source models on simpler tasks but not harder tasks.*
>
> In the second-to-last paragraph of the “contributions” in the Introduction, we have highlighted the findings we believe may be of the greatest interest to the reader. We believe each finding may impact different readers in various ways. You can find the detailed findings in Section 4.

---

> ### Author Response · Authors · 2024-11-14
>
> 4. *Why is all API-based agents the authors evaluate based on the ReACT framework, it would be good if they could provide additional evaluation/analysis on other frameworks, such as CodeAct.*
>
> ReACT is a generate-feedback alternating execution method, and our reasons for using the ReACT are as follows:
>
> - It aligns with the settings of prior work, such as MetaTool, ToolLLM, ToolAlpaca, and ToolQA mentioned in Table 2, whereas APIBench, APIBank, and ToolBench use zero-shot and multi-shot settings.
> - A simple yet effective framework better reflects the model’s capabilities, rather than the capabilities of the framework itself.
>
> Our research focuses on evaluating API-based agents, whereas CodeAct—mentioned in your feedback—is a valuable approach for generating Python code, as referenced in the second paragraph of our related work. However, it is not directly relevant to our research.
>
> We will include a discussion of CodeAct in the related works section.

---

> ### Author Response · Authors · 2024-11-14
>
> Thank you once again for your feedback. We truly hope that our responses above have helped clarify any concerns you may have. We would be very grateful if you could carefully consider your final rating. Please don’t hesitate to reach out with any additional questions or comments—I would be more than happy to assist further.

---

> ### Author Response · Authors · 2024-11-19
>
> Hello,
>
> The issues you raised include:
>
> Weakness 1: Concerns about the clarity of certain statements in the paper. We sincerely apologize for any confusion caused and hope you can understand our choices. ShortcutsBench explores an innovative perspective by constructing a pipeline to extract high-quality data from existing Digital Automation Platforms for benchmarking. Inevitably, this involves specific terminology and expressions related to these platforms. While we have made efforts to simplify the language, add footnotes where ambiguity might arise, and include detailed processes in the appendix, we acknowledge that we may not have conveyed every statement as clearly as possible.
>
> Weakness 2: Concerns about the frequency of certain citations. We regret any inconvenience caused. Our intention in using multiple references across different sections was to ensure that every claim in the paper is properly supported by relevant paper, and we hope for your understanding in this regard.
>
> Regarding your question about the use of the ReACT framework, please refer to our detailed response for clarification.
>
> ICLR emphasizes innovative ideas, rigorous work, and full openness. To our best known, ShortcutsBench represents the first attempt to leverage data from Digital Automation Platforms to construct high-quality benchmarking datasets. To ensure robustness, we conducted experiments on agents constructed using 10 open-source and proprietary models, while maintaining full openness in data acquisition, processing, and evaluation, with comprehensive logs throughout the process.
>
> We greatly appreciate your evaluation of the paper’s Soundness (3/4) and Contribution (3/4). We hope our responses can address your concerns regarding Presentation (1/4). If you have more specific feedback on Presentation, we welcome your suggestions. We sincerely hope you will take all aspects of ShortcutsBench into consideration when determining your final score.
>
> Thank you again for your thoughtful feedback.

---

> > ### Comment · Reviewer_AydN · 2024-11-24
> >
> > Thank you for your responses. Your comments on points 1, 2, 3, and 4 addressed my concerns and clarified the ambiguities in your paper. However, I believe that ICLR allows authors to submit a revised version of their paper during the rebuttal phase. For the revisions you have promised to include in the final version, as well as the additional explanations to improve the paper's readability and accessibility for readers, it would be beneficial to incorporate them into a revised submission. Since the current presentation of your paper has not improved, I will maintain my current score for now. **If you submit a revised version with the promised additions highlighted in the revised version, I will raise my score.**

---

> ### Author Response · Authors · 2024-11-24
>
> Thank you so much for your prompt response and positive feedback! We have carefully reviewed and incorporated all your comments and concerns into the revised [PDF](https://openreview.net/pdf?id=kKILfPkhSz).
>
> Specifically:
>
> ### Concern 1:
>
> We made substantial revisions to Section 3, aiming to better convey our intentions more precisely. Key updates include:
>
> - **Redrawing Figure 1**:
>     - Removed unnecessary icons (e.g., the spider icon representing crawlers).
>     - Adjusted the placement of critical steps (e.g., “Deduplication”) for greater accuracy.
>     - Refined the depiction of app sources, added a legend, and introduced Step 6 to indicate data filtering.
> - **Reorganizing Section 3.1**:
>     - Removed references to most URLs.
>     - Expanded on *“after deduplicating based on iCloud links, …”*.
>     - Reorganized the discussion of app sources, relocating details about “WFActions.json” to the second paragraph.
>     - Further elaborated on API deduplication in the second paragraph. For a detailed discussion on why we perform API deduplication, please refer to Appendix A.1.
>     - In the final paragraph, we remind readers to refer to Appendix A.1 for further details on data acquisition.
> - **Streamlining Section 3.2**: Simplified descriptions of **APIs** for improved readability.
> - **Adding Context in Section 3.2** (Query and action sequence):
>     - Included descriptions of efforts to maintain shortcut usability in the first paragraph.
>     - Highlighted efforts to ensure query accuracy in the last paragraph.
> - **Refining Section 3.3**: Simplified citations and phrasing for clarity.
> - Additional edits for smoothness and coherence throughout the section.
>
> ### Concern 2:
>
> We substantially reduced and refined citations across the paper. Previously, we aimed for comprehensive references for each point, but now we’ve retained only the most relevant citations:
>
> **1 Introduction:**
>
> Significantly trimmed references, especially in the second paragraph (“Some existing benchmarks/datasets…”). Initially, every critique of prior work was accompanied by specific citations. Now, we have removed most of these references and instead direct readers to Table 2 for a concise summary.
>
> **2 Related work & 3 Dataset:**
>
> Significantly streamlined references: we ensure that each section contains no more than two citations, with the same paper cited only once per paragraph.
>
> **4 Evaluation**
>
> All references to LLMs have been removed, and citations for related work have been streamlined. We believe the paper provides a sufficiently precise description of the models, enabling readers to independently locate and utilize them without citation.
>
> ### Concern 3:
>
> We relocated the findings you highlighted to the “key contributions” part at the end of the introduction.
>
> ### Concern 4:
>
> In Section 4.1 (“Prompt Template”), we added citations for the ReACT method under “Following existing work” while limiting the overall number of references for brevity.
>
> ### Additional Notes:
>
> We understand that the detailed process of collecting, cleaning, and constructing ShortcutsBench cannot be fully covered within the constraints of a 10-page paper. Thus, we focused on ensuring the content is both concise and clear. For more detailed explanations, please refer to the Appendix, where we revised the descriptions of the entire collection, cleaning, and construction process. Additionally, the implementation details and logs are all available at https://anonymous.4open.science/r/ShortcutsBench.
>
> In addition to addressing your feedback, we have also made revisions to various parts of the paper based on comments from other reviewers. You may refer to the specific comments for details.
>
> As the discussion phase concludes on November 26 (AOE), we have implemented all the changes we committed to so far. However, there may still be some oversights, and we will use the remaining time to carefully review and refine our revisions.
>
> Thanks again! We greatly appreciate your feedback and look forward to your further comments! If you have any questions further, please feel free to reach out—I’m happy to assist.

---

> > ### Comment · Reviewer_AydN · 2024-11-26
> >
> > Thank you for your fast response. I increased my scores.

---

> > > ### Author Response · Authors · 2024-11-29
> > >
> > > Thank you for your response! We have conducted additional experiments on language models fine-tuned specifically for Agents, hoping to provide further insights. If you’re interested, feel free to check out the replies in XY2i’s thread.

---

### Official Review · Reviewer_XY2i · 2024-11-08

**Soundness:** 4
**Presentation:** 4
**Contribution:** 3
**Rating:** 8
**Confidence:** 4

**Summary:**

This paper introduces a comprehensive, large-scale benchmark designed to evaluate API-based agents in real-world scenarios. It addresses the shortcomings of existing benchmarks and datasets, which often lack the richness and complexity needed to thoroughly assess different LLM-based agent models. To overcome these limitations, the authors have developed a high-quality benchmark encompassing 88 applications, 1,400 APIs, and an average of 21 actions per API, each covering various aspects of real-world deployments.

In the evaluation, the authors tested 10 LLMs on several key tasks: API selection, parameter filling, and the models' ability to recognize when to request additional input from either the system or the user. The experimental results show interesting insights, such as the challenges these models face with multi-step reasoning and understanding when external input is necessary.

**Strengths:**

This paper makes a notable contribution by creating a comprehensive benchmark for API-based agents, utilizing data extracted from Shortcuts. Compared to other API-based benchmarks, it offers several benefits, including a focus on the agents' ability to request necessary input from either the assistant or user and diverse difficulty of tasks. It covers a range of tasks, from simple ones to those involving complex APIs, queries, and action sequences. Additionally, the paper ensures quality by involving human verification, with shortcut developers serving as annotators.

The evaluations demonstrate insightful findings, especially regarding the challenges agent models face in reasoning and planning capabilities as indicated by API selection, as well as the difficulties weaker LLMs encounter in API parameter filling. These insights are valuable for the development of more advanced agent models.

**Weaknesses:**

1. Although the paper emphasizes that the benchmark includes high-quality human-annotated action sequences from shortcut developers and queries derived from real user demands, it only mentions the shortcut developers are our annotators. Further details in this area would be beneficial.

2. In section 3.2, the paper describes using GPT-4o to simulate user queries. However, it would be helpful to include the steps taken to verify the correctness and ensure the diversity of these user queries.

3. The evaluation primarily features selected proprietary models like GPT-4o and Gemini. For open-source LLMs, it mainly compares with general LLMs such as Qwen-2-70B and LLaMA-3-70B. Considering comparisons with more robust, specifically developed AI Agent models such as AgentLM (70B from https://github.com/THUDM/AgentTuning) and xLAM (8x7b or 8x22b from https://github.com/SalesforceAIResearch/xLAM) could provide more insights.

4. The source of the reported numbers in Table 1 could be more clearly specified. There is uncertainty regarding whether models like Qwen-2.5-7B and LLaMA-3-8B consistently achieve more than 90% on ToolBench and over 80% on ToolLLM. Furthermore, given that MetaTool (https://github.com/HowieHwong/MetaTool) does not provide comprehensive details or code for model evaluation and metrics, more information is necessary to verify the accuracy of the table data.

**Questions:**

Please refer comments in above fields.

---

> ### Author Response · Authors · 2024-11-14
>
> 1. *Although the paper emphasizes that the benchmark includes high-quality human-annotated action sequences from shortcut developers and queries derived from real user demands, it only mentions the shortcut developers are our annotators. Further details in this area would be beneficial.*
>
>     In addition to the inherent quality assurance of shortcuts developed by shortcut developers in ShortcutsBench, we implemented **three** additional steps to ensure the quality of action sequences:
>
>     1. As mentioned in the final paragraph of Section 3.1 (and detailed in Appendix A.1), we removed duplicate shortcuts by iCloud links. Specifically, we removed duplicate shortcuts based on their iCloud links and action sequences; any shortcuts with either of these identical elements were deduplicated.
>     2. Additionally, we manually filtered out shortcuts that relied on outdated or inactive APIs, as well as those requiring APIs from paid apps. This ensured that all included shortcuts use operational APIs. You can review the details of this process in our [open-source documentation](https://anonymous.4open.science/r/ShortcutsBench/deves_dataset/dataset_src_valid_apis/README.md). This was a manual process, involving two reviewers who tested on macOS Sonoma and iOS 17, downloading all relevant apps and verifying API functionality. APIs deemed obsolete were removed directly; you can view a list of these APIs [here](https://anonymous.4open.science/r/ShortcutsBench/deves_dataset/dataset_src_valid_apis/APIs_to_be_filtered_out.py). Apps used in this evaluating process can be downloaded following the guidance in our [README](https://anonymous.4open.science/r/ShortcutsBench/deves_dataset/dataset_src_valid_apis/README.md).
>     3. Finally, we signed all remaining shortcut files, as described in [ShortcutsBench documentation](https://anonymous.4open.science/r/ShortcutsBench/README.md), and loaded them into the macOS Sonoma Shortcuts app, where unsigned shortcuts cannot be imported. We tested these shortcuts by running them individually, as loading thousands of shortcuts at once would cause the app to crash. To facilitate evaluating on macOS Sonoma, we added the required apps to the Applications folder, allowing the Shortcuts app to capture and execute the corresponding shortcut actions. Note that some iOS-only apps require specific modifications (known as [Clutch](https://github.com/KJCracks/Clutch)) to run on macOS. Throughout this process, we manually excluded empty or low-quality shortcuts, resulting in the final dataset of 7,627 shortcuts as described in our paper.
>
>     For details on how we ensured the quality of queries, we combine the response to this question with the Question 2 you mentioned.
>
>     If you have any further questions regarding the details, please feel free to ask. We would be happy to assist you.

---

> ### Author Response · Authors · 2024-11-14
>
> 2. *In section 3.2, the paper describes using GPT-4o to simulate user queries. However, it would be helpful to include the steps taken to verify the correctness and ensure the diversity of these user queries.*
>
> To ensure the accuracy of our constructed queries, we employ two main strategies:
>
> 1. As detailed in Section 3.2, we leverage a more advanced model, GPT-4o, for query construction. GPT-4o was selected following a preliminary experiment comparing GPT-3.5 (active at the time of evaluating), GPT-4o, and Gemini-1.5-Pro across 100 queries. Human evaluators scored GPT-4o-generated queries highest, outperforming the other two models. GPT-4o effectively captures required parameters and provides a clear query description, meeting our criteria in 94 out of 100 cases. In contrast, GPT-3.5 often missed specific parameters, and Gemini-1.5-Pro performed slightly below GPT-4o.
>
>     Although we cannot guarantee all 7,627 queries precisely meet our requirements, we consider our approach reasonable. Similar work, such as ToolLLM, was published in ICLR '24 with Spotlight, using GPT for extensive query and action sequence generation without guaranteeing full accuracy.
>
> 2. As described in Section 3.2, our query data includes a range of information—not only API data but also action sequences and functional descriptions provided by humans for each shortcut command. For the 100 queries generated, 94 met our manual verification criteria, requiring that the query adequately describes the shortcut’s function and integrates both primitive and enum data type parameters necessary for API calls. These parameters were extracted from the full action sequences, and the template we used is available in Appendix A.2.
>
> Regarding diversity, we did not actively introduce diversity into our queries; instead, the diversity of our dataset derives from the shortcuts themselves. Each shortcut corresponds to a unique constructed query. As mentioned in “*Further Processing”* in Section 3.3 (or Appendix A.3), we categorized our shortcuts into 8 categories, following the classification method used in the Apple App Store, which further reflects the diversity of our dataset.
>
> If you have any additional questions about the details, please don't hesitate to reach out. We would be glad to provide further clarification.

---

> ### Author Response · Authors · 2024-11-14
>
> 3. *The evaluation primarily features selected proprietary models like GPT-4o and Gemini. For open-source LLMs, it mainly compares with general LLMs such as Qwen-2-70B and LLaMA-3-70B. Considering comparisons with more robust, specifically developed AI Agent models such as AgentLM (70B from https://github.com/THUDM/AgentTuning) and xLAM (8x7b or 8x22b from https://github.com/SalesforceAIResearch/xLAM) could provide more insights.*
>
> In our experimental design, we categorized the LLMs to be evaluated from both closed-source and open-source perspectives. Within each category, we selected models that are considered to have varying reasoning capabilities, which we believe is a reasonable classification. Moreover, the LLMs we evaluated surpass existing works in both quantity and diversity, as shown in Table 2:
>
> | Tool | Number of Large Language Models (LLMs) |
> | --- | --- |
> | MetaTool (ICLR’24) | 8 LLMs |
> | ToolLLM (ICLR’24) | 5 LLMs |
> | APIBench (NeurIPS’24) | 4 LLMs |
> | ToolAlpaca (arXiv’23) | 5 LLMs |
> | APIBank (EMNLP’23) | 6 LLMs |
> | ToolBench (arXiv’24) | 4 LLMs |
> | ToolQA (NeurIPS’23) | 2 LLMs |
> | ToolLens (CIKM’24) | - |
>
> Moreover, closed-source language models such as Gemini-1.5-Pro, Deepseek-2-Chat, Deepseek-2-Coder, Gemini-1.5-Flash, GPT-4o-Mini, and ChatGLM-4-Air we used have been specifically fine-tuned for function calling.
>
> ShortcutsBench differs significantly from AgentLM/xLAM. The goal of ShortcutsBench is to provide a benchmark that is sufficiently challenging, rich, and distinctive, aligning more closely with [AgentBench](https://arxiv.org/pdf/2308.03688) and [MetaTool](https://arxiv.org/abs/2310.03128). I believe the experiments conducted within ShortcutsBench have effectively validated the problems mentioned in the Introduction. In contrast, AgentLM and xLAM focus on specialized fine-tuning of language models to enhance their API-calling capabilities in specific domains. It is entirely feasible to use the APIs provided by ShortcutsBench in conjunction with the data generation methods of AgentLM/xLAM to generate data that improves a language model’s performance in specific areas.
>
> Despite this, we have fully accounted for the potential evaluation and analysis of additional models. Once relevant models are deployed, evaluating results can be obtained with a single command using Python's `openai` package. Detailed instructions on how to run our code for new model evaluations are provided in our code and data repository, accessible via this link: https://anonymous.4open.science/r/ShortcutsBench/experiments/README.md. You only need to make minor modifications to a bash script, such as https://anonymous.4open.science/r/ShortcutsBench/experiments/restart_all_experiments_llama_3_72b.sh, and add the model in https://anonymous.4open.science/r/ShortcutsBench/experiments/all_experiments.py. This will allow you to conduct the evaluation with one command. Our code is designed to handle network errors automatically, reattempting execution until all results are successfully obtained.
>
> If you have any other questions or need further clarification on the details, please feel free to let us know. We’d be happy to assist you.

---

> ### Author Response · Authors · 2024-11-14
>
> 4. *The source of the reported numbers in Table 1 could be more clearly specified. There is uncertainty regarding whether models like Qwen-2.5-7B and LLaMA-3-8B consistently achieve more than 90% on ToolBench and over 80% on ToolLLM. Furthermore, given that MetaTool (https://github.com/HowieHwong/MetaTool) does not provide comprehensive details or code for model evaluation and metrics, more information is necessary to verify the accuracy of the table data.*
>
> In the second paragraph of the Introduction, we mention, “Our evaluation of these less intelligent LLMs on 3 representative existing benchmarks/datasets,” meaning that the results in Table 1 are from our own evaluation. We evaluated the API selection performance on 3 representative datasets across various smaller-scale language models, which are generally considered to have lower reasoning capabilities. All data, code, and results used in these tests are fully open-sourced and available at https://anonymous.4open.science/r/ShortcutsBench/pre_experiments/README.md, where you can review them to verify the source of the results.
>
> Regarding your comment that “*MetaTool does not provide comprehensive details or code for model evaluation and metrics”,* this does not impact our experiments. We used https://github.com/HowieHwong/MetaTool/blob/master/dataset/plugin_des.json as the tool description and selected https://github.com/HowieHwong/MetaTool/blob/master/dataset/data/all_clean_data.csv as the query data, evaluating only the accuracy of API selection.
>
> Furthermore, to ensure a fair comparison, we used the same prompt template for testing as described in Section 4.

---

> ### Author Response · Authors · 2024-11-14
>
> Thank you once again for your valuable feedback and insights. I truly hope that our responses above have helped clarify any concerns you may have. We would be very grateful if you could carefully consider your final rating (out of 10). Please don’t hesitate to reach out with any additional questions or comments—I would be more than happy to assist further.

---

> ### Author Response · Authors · 2024-11-19
>
> Your concerns regarding Weaknesses 1 and 2, which primarily focus on the dataset construction process, have been carefully considered. We have implemented three additional steps to ensure the quality of action sequences and two additional steps to enhance the quality of user queries. For detailed explanations, please refer to our specific responses. To ensure transparency and accessibility, we have made all processes, including methods, details, and logs, available at: https://anonymous.4open.science/r/ShortcutsBench/.
>
> For Weakness 4, regarding the source of results in Table 1, we have open-sourced the complete dataset, methods, and operational instructions used during testing at: https://anonymous.4open.science/r/ShortcutsBench/pre_experiments/. You are welcome to review the relevant results or reproduce them yourself. For Weakness 3, concerning additional experiments on 10 Agents, please refer to our detailed responses.
>
> We sincerely appreciate your positive evaluation of the paper’s Soundness (4/4), Presentation (4/4), and Contribution (3/4). ShortcutsBench approaches the topic from a previously unexplored perspective, constructing a pipeline to extract high-quality data from existing Digital Automation Platforms as a benchmark dataset. This benchmark represents a challenging end-to-end process for API-based agent tasks, including API selection, parameter filling, and awareness for requesting missing information. It also uncovers many intriguing findings about API-based agents that we believe add significant value.
>
> If you have any further concerns that might influence the final score, please feel free to share them. We would be more than happy to provide additional clarification.

---

> ### Author Response · Authors · 2024-11-24
>
> Thank you for your valuable feedback and insights. I hope our responses have addressed your concerns. With the Discussion Period ending on November 26, I am eager to ensure all your questions are fully resolved. We would greatly appreciate it if you could consider your final rating (out of 10). Please don’t hesitate to reach out with any further questions or comments—I’m here and ready to assist.

---

> ### Author Response · Authors · 2024-11-24
>
> Thank you for your feedback. We have carefully addressed all your comments in the revised [PDF](https://openreview.net/pdf?id=kKILfPkhSz) and look forward to hearing your response.
>
> Specifically:
>
> ### Concern 1 & Concern 2:
>
> In Section 3.2 (Query and action sequence), we added descriptions of our efforts to ensure the quality of queries, with more detailed explanations provided in Appendices A.1 and A.2:
>
> - Descriptions of efforts to maintain shortcut usability are included in the first paragraph.
> - Efforts to ensure query accuracy are highlighted in the final paragraph.
>
> You can also review the complete processing logs and records, which are open-sourced at https://anonymous.4open.science/r/ShortcutsBench.
>
> ### Concern 3:
>
> We added an introduction and comparison of AgentLM and xLAM in Section 4.1:
>
> “We did not compare with specialized API-calling fine-tuned LLMs like AgentLM or xLAM for two reasons. First, our selected models already cover a range of intelligence levels, including closed-source models fine-tuned for API-calling tasks. Second, our goal is to provide a benchmark that is challenging, rich, and distinctive, which has been validated under the current setup. While AgentLM and xLAM focus on fine-tuning LLMs for API usage in specific domains, the APIs and methods in ShortcutsBench could be combined with their approaches to generate data for enhancing performance in targeted areas.”
>
> ### Concern 4:
>
> The sentence in the second paragraph of the Introduction, “Our evaluation on API selection of these less intelligent LLMs…,” indicates that this evaluation was conducted by us. You can further explore the fully open-sourced materials at https://anonymous.4open.science/r/ShortcutsBench/pre_experiments/README.md to review the experimental results.
>
> ### Additional Notes:
>
> As the discussion phase concludes on November 26 (AOE), we have implemented all the changes we committed to so far. However, there may still be some oversights, and we will use the remaining time to carefully review and refine our revisions.
>
> We greatly appreciate your high evaluation of the paper’s Soundness (4/4), Presentation (4/4), and Contribution (3/4). ShortcutsBench offers a novel perspective, creating a benchmark dataset from Digital Automation Platforms to evaluate API-based agents on tasks like API selection, parameter filling, and handling missing information. It also reveals valuable insights into these agents’ performance.
>
> We hope you will consider the contributions of ShortcutsBench when providing your final score. Please don’t hesitate to reach out with any questions—I’d be happy to assist.

---

> > ### Comment · Reviewer_XY2i · 2024-11-24
> > **Official comments by Reviewer**
> >
> > Thank you for your thorough responses to my questions. Regarding Table 2 and the selection of LLMs, it seems that the table lists the number of LLMs rather than the number of unique evaluated LLMs. Additionally, models such as AgentLM, xLAM, and others like Lemur (https://arxiv.org/pdf/2310.06830) are strong open-source models specifically designed and trained for AI Agent tasks. These models differ from closed-world LLMs and general LLMs like LLAMA and QWEN. Since the paper aims to provide a agent benchmark that is challenging, rich, and distinctive, which has been validated under the current setup, it would be more convincing to include benchmarking results for these robust Agent models.

---

> > > ### Author Response · Authors · 2024-11-29
> > >
> > > Thank you for your response!
> > >
> > > Based on your suggestions, we have added experiments involving **11** LLMs, including **6** LLMs fine-tuned specifically for agent scenarios, and **5** LLMs as their base model to evaluate the applicability of fine-tuning for API-based agents. These include **2 xLAM** series of models, **3 agentlm** series of models, and **1 lemur** model.
> > >
> > > The **AgentLM** series model is **fully fine-tuned** on a dataset of **1,866** traces from simple code data. **Lemur** is fine-tuned based on scripting or interpreted languages of [**The Stack**](https://huggingface.co/datasets/bigcode/the-stack), a dataset of code from GitHub. The **xLAM** series, on the other hand, is a **specialized model fine-tuned for API-based agents**, with a predefined format for API calls.
> > >
> > > Additionally, to explore whether these fine-tunings for agent scenarios are applicable to API-based agents, we conducted experiments on the base models of these fine-tuned models. These include the **Mistral-7B (xLAM-7b-r)** and **Mixtral-8x7B (xLAM-8x7B-r)** as well as the **LLaMA-2-7B (agentlm-7b)**, **LLaMA-2-13B (agentlm-13b)**, and **LLaMA-2-70B (agentlm-70b),** resulting in **5** base models.
> > >
> > > The links to all the LLMs we evaluated are as follows:
> > >
> > > - **6** LLMs fine-tuned specifically for agent scenarios:
> > >     - https://huggingface.co/THUDM/agentlm-7b
> > >     - https://huggingface.co/THUDM/agentlm-13b
> > >     - https://huggingface.co/THUDM/agentlm-70b
> > >     - https://huggingface.co/Salesforce/xLAM-7b-r
> > >     - https://huggingface.co/Salesforce/xLAM-8x7b-r
> > >     - https://huggingface.co/OpenLemur/lemur-70b-chat-v1
> > > - **5** base LLMs:
> > >     - https://huggingface.co/meta-llama/Llama-2-7b-chat-hf
> > >     - [https://huggingface.co/meta-llama/Llama-2-13b-chat-hf](https://huggingface.co/meta-llama/Llama-2-7b-chat-hf)
> > >     - [https://huggingface.co/meta-llama/Llama-2-70b-chat-hf](https://huggingface.co/meta-llama/Llama-2-7b-chat-hf)
> > >     - https://huggingface.co/mistralai/Mistral-7B-Instruct-v0.1
> > >     - [https://huggingface.co/mistralai/Mixtral-8x7B-Instruct-v0.1](https://huggingface.co/mistralai/Mistral-7B-Instruct-v0.1)

---

> > > ### Author Response · Authors · 2024-11-30
> > >
> > > Thank you for sharing your valuable feedback and insights. We hope our responses have effectively addressed your concerns. As the Discussion Period concluded on December 2 (AoE), I want to ensure that all your questions are thoroughly resolved. We would greatly appreciate it if you could consider your final rating (out of 10). Should you have any additional questions or comments, please feel free to reach out—I’m here and happy to assist.

---

> > > ### Author Response · Authors · 2024-12-02
> > >
> > > As the discussion period ends on December 2 (AoE), we are eager to ensure that all the questions have been thoroughly resolved. We hope that our responses have adequately addressed your concerns. Your feedback is invaluable to us, and we would greatly appreciate it if you could take a moment to provide a final rating and feedback.

---

> ### Author Response · Authors · 2024-11-29
>
> We have placed the results of the 6 LLMs fine-tuned for agent scenarios alongside their corresponding base models in the same cell of the table for easy comparison. If the fine-tuned model outperforms the original, the corresponding result is highlighted in bold.
>
> The details of how we processed the experimental results are as follows:
>
> - We used **gpt-4o-mini** to convert outputs with non-standard formats into our standardized format for evaluation. We manually verified **50** randomly selected samples and confirmed that all conversions were correct. The conversion script can be found at: https://anonymous.4open.science/r/ShortcutsBench/data_for_agent_llm/reformat_output_2_correct_format.py. The reason for this processing step is that we observed poor instruction-following capabilities in **llama-2-7b**, **llama-2-13b**, **agentlm-7b**, **and agentlm-13b** (as described is Section setup in the initial version of ShortcutsBench), particularly when it came to generating outputs in the specific formats required by the prompts. For **llama-2-7b**, **llama-2-13b**, **agentlm-7b**, and **agentlm-13b**, this limitation may arise directly from the models’ inherent deficiencies in instruction-following capabilities. Incorrect output formats can lead to errors in result parsing, ultimately underestimating the experimental results.
> - Since the context lengths of **llama-2-7b**, **llama-2-13b**, **llama-2-70b**, **agentlm-7b**, **agentlm-13b**, **agentlm-70b**, and **lemur-70b-chat-v1** are limited to **4k** tokens, it is challenging for these models to handle tasks in the more difficult range of (15,30]. Therefore, we only report results for the first **3** difficulty levels.
> - The models we evaluated include several 70B models, which require 257GB of storage and long time for download. Additionally, these LLMs, fine-tuned for agent scenarios, do not have publicly available APIs. Even with models like Llama-2, there are few callable APIs, so we had to deploy them ourselves. Running 70B models demands significant resources, so we spent much time on the setup.
>
> The detailed experimental results have been updated at: https://drive.google.com/drive/folders/1kYkqTWrukdb2nOMuK-O5TZo_cJ7SVF3E?usp=sharing
>
> Files without the `.after_format` suffix represent the outputs prior to formatting.

---

> ### Author Response · Authors · 2024-11-29
>
> | **Model**                         | **Llama-2-7b / agentlm-7b** | **Llama-2-13b / agentlm-13b** | **Llama-2-70b / agentlm-70b** | **Mistral-7b / xLAM-7b-r** | **Mixtral-8x7b / xLAM-8x7b-r** | **Llama-2-70b / lemur-70b-chat-v1** |
> | --------------------------------- | --------------------------- | ----------------------------- | ----------------------------- | -------------------------- | ------------------------------ | ----------------------------------- |
> | **Context Length**                | **4k**                      | **4k**                        | **4k**                        | **32k**                    | **32k**                        | **4k**                              |
> | **API Selection**                 | **32.75** / 24.95           | **38.59** / 30.57             | **52.13** / 43.41             | 17.03 / **70.56**          | 70.21 / **85.29**              | 52.13 / **52.45**                   |
> |                                   | 12.57 / **17.43**           | 25.99 / **35.48**             | **43.08** / 37.34             | 3.27 / **50.82**           | 49.15 / **51.72**              | 43.08 / **47.08**                   |
> |                                   | 23.70 / **32.78**           | 35.45 / **40.78**             | **57.76** / 39.22             | 0.50 / **66.33**           | 29.92 / **29.67**              | 57.76 / **60.39**                   |
> | **API Parameter Value Filling**   | **51.11** / 34.88           | 30.77 / **31.82**             | **72.86** / 52.83             | 40.00 / **66.67**          | 53.45 / **68.93**              | **72.86** / 45.45                   |
> |                                   | 32.14 / **35.71**           | 12.50 / **40.98**             | **45.05** / 37.31             | 40.54 / **59.32**          | 33.72 / **46.76**              | **45.05** / 36.49                   |
> |                                   | 0.00 / 0.00                 | 11.11 / **40.00**             | 6.67 / **11.11**              | 11.11 / **31.82**          | 27.20 / **42.71**              | 6.67 / **26.92**                    |
> | **Recognition of Need for Input** | 7.41 / **10.53**            | 5.00 / **5.26**               | **21.74** / 0.00              | 0.00 / **5.56**            | 6.25 / **11.90**               | **21.74** / 16.00                   |
> |                                   | **12.50** / 4.17            | 0.00 / **3.57**               | 7.69 / **9.09**               | 0.00 / **2.56**            | 19.27 / **19.91**              | **7.69** / 3.57                     |
> |                                   | 0.00 / 0.00                 | 0.00 / 0.00                   | 0.00 / 0.00                   | 0.00 / 0.00                | 0.00 / 0.00                    | 0.00 / 0.00                         |
>
> -
>
> - Performance on **Recognition of Need for Input:**
>
>   - **All evaluated LLMs performed exceptionally poorly in recognition of need input**, significantly lagging behind the General LLMs mentioned in the paper of ShortcutsBench.
>   - **Compared to the performance of the base LLMs** before fine-tuning, **neither** the AgentLM and Lemur series **models fine-tuned** on code nor the xLAM series models fine-tuned specifically for API-based agents **showed any improvement** in this aspect. This suggests the need for targeted fine-tuning specifically aimed at this capability.
>
> ##

---

> > ### Author Response · Authors · 2024-11-29
> >
> > - Performance on **API Parameter Value Filling**:
> >   - **The evaluated LLMs perform significantly worse in API Parameter Value Filling compared to the General LLMs mentioned in the ShortcutsBench.** The General LLMs achieves accuracies of 94.29/83.92/75.53 across three difficulty levels, whereas the best results from these evaluated models only reach accuracies of 72.86/59.32/42.71.
> >   - Compared to the base LLMs before fine-tuning, **the Code-based agents**, including the AgentLM and Lemur models fine-tuned on code, **did not achieve a general improvement in parameter-filling accuracy for API-based tasks** and even exhibited performance degradation. In contrast, **the API-based agents**, such as the xLAM series fine-tuned specifically for API-based agents, **showed a consistent improvement in parameter-filling accuracy**.
> >   - The exitsing fine-tuning methods for API-based agents (xLAM-7b-r / xLAM-8x7b-r) are effective for parameter filling in short sequences, but **accuracy drops sharply as the context length increases**.
> > - Performance on **API selection**:
> >   - **Methods** fine-tuned specifically **for API-based agents**, such as the xLAM series models, can significantly enhance agents’ API selection accuracy. The xLAM series achieves reasonable accuracy in API selection, even compared to general LLMs.
> >   - Compared to xLAM-7b-r, the API selection accuracy of xLAM-8x7b-r is significantly lower, **suggesting** that **existing fine-tuning methods specialized for agents may only achieve notable improvements on models with relatively weaker foundational capabilities**.
> >   - **Methods fine-tuned on code can still enhance agents’ API selection accuracy**, as agentlm-7b/13b and lemur-70b-chat-v1 shows.
> >   - The performance drop of agentlm-70b may be attributed to the fine-tuning dataset used for the agentlm series lacking realism and complexity.
> >
> >       An example of the data used for AgentLM is as follows. For more details, see [https://huggingface.co/datasets/THUDM/AgentInstruct](https://huggingface.co/datasets/THUDM/AgentInstruct). The format of the first example is as follows.
> >
> >       ```
> >       tell me how many files are in the directory "/etc"?
> >
> >       From: gpt
> >       Loss: false
> >       Value:
> >
> >       Think: To count the files in /etc, I need to print all the files in it.
> >       Act: bash
> >       bash
> >       ls /etc
> >
> >       From: human
> >       Loss: null
> >       Value: The output of the OS:
> >       cpi cron.hourly fuse.conf iproute2 lvm networkd-dispatcher protocols selinux tmpfiles.d [truncated because the output is too long]
> >
> >       From: gpt
> >       Loss: false
> >       Value:
> >       Think: The output has been truncated because it is too long, so I need to count files by script directly.
> >       Act: bash
> >       bash
> >       ls -1 /etc | wc -l
> >
> >       From: human
> >       Loss: null
> >       Value: The output of the OS: 220
> >
> >       From: gpt
> >       Loss: false
> >       Value:
> >       Think: Now I get the answer, it is 220.
> >       Act: answer(220)
> >       ```
> >
> > ##

---

> > > ### Comment · Reviewer_XY2i · 2024-12-02
> > > **Official comment by Reviewer**
> > >
> > > Thank you for your detailed response and the newly added evaluation results and analysis for the three agent models. They are highly valuable and insightful for the community. Could you update the revised version with these new details so I can adjust my score to fall within the acceptance range?

---

> > > > ### Author Response · Authors · 2024-12-03
> > > >
> > > > Thank you for your response. With **only a few hours left before the discussion deadline**, we earnestly request that you provide your final score before the deadline.
> > > >
> > > > We hope you understand that, in addition to the experiments on these 11 additional models, **all other concerns have been addressed in the revised PDF**.
> > > >
> > > > However, **as the deadline for modifying the PDF (November 27th) has passed, we are unable to make further changes at this time.**
> > > >
> > > > Nonetheless, **we commit to: once the option to modify the PDF becomes available, we will make every effort to integrate the results into both the main text and the appendix**, with clear and prominent markings to indicate their location.
> > > >
> > > > **All comments from openreview are publicly visible, and we will adhere to our commitment.**
> > > >
> > > > We look forward to receiving your final response as soon as possible.

---

> > > > > ### Comment · Reviewer_XY2i · 2024-12-03
> > > > > **Official comment from Reviewer**
> > > > >
> > > > > Thanks for your commitment. Based on the current content of the work, I have raised my score to 8.

---

> > > > > > ### Author Response · Authors · 2024-12-03
> > > > > >
> > > > > > Thank you for your affirmation! We assure you that as soon as it is permitted to modify the PDF, we will promptly update it to incorporate the current results. Wishing you all the best!

---

> ### Author Response · Authors · 2024-11-29
>
> We hope our experiments can address your concerns! If you have any further questions, please don’t hesitate to reach out to us. We would be delighted to assist you and look forward to hearing from you!

---

> ### Author Response · Authors · 2024-12-01
> **(1 day left) Follow-up on Remaining Concern 3 and Request for Final Feedback**
>
> We have carefully addressed the remaining Concern 3 and look forward to your feedback. With the Discussion Period ending on December 2 (AoE) and only one day remaining, We want to ensure that all your questions have been fully resolved. We eagerly await your response.

---

> ### Author Response · Authors · 2024-12-02
>
> As the Discussion Period concludes today December 2 (AoE), we are eager to ensure that all questions have been thoroughly addressed. We hope our responses have sufficiently resolved your concerns. Your feedback is invaluable to us, and we would greatly appreciate it if you could take a moment to provide a final rating and share your thoughts.

---

> ### Author Response · Authors · 2024-12-03
>
> Thank you for your response! We’re delighted to have your approval. Unfortunately, as the deadline for modifying the PDF has passed (November 27), we are unable to make any changes or re-upload it at this time. However, we assure you that once the option to modify the PDF becomes available, we will make every effort to integrate the results into both the main text and the appendix, with clear and prominent markings to indicate their location.

---

### Author Response · Authors · 2024-11-14

We thank all reviewers for their comments and feedback.

We summarized the strengths highlighted and categorized the concerns as follows:

The reviewer has affirmed the benchmark, method, evaluation, findings, and artifacts of our paper.
1. **benchmark**: notable contribution by creating a comprehensive benchmark (XY2i);  more holistic benchmark that contains real APIs, well-designed queries and actions (AydN);
2. **method**: interesting approach (2zSS); this benchmark seems to be scalability (2zSS);
3. **evaluation**: detailed analysis (AydN); comprehensive evaluation (2zSS);
4. **findings**: insightful findings (XY2i); analysis results in 4.2 are insightful (vqjw);
5. **open-source**: the artifacts are provided and well-organized (vqjw);

The reviewer expressed concerns about the dataset construction process, raised questions regarding the methodological details, and suggested that the authors conduct additional experiments in certain areas:

1. **concerns about the dataset construction process**: Further details about action sequences would be beneficial (XY2i); including steps taken to verify the correctness and ensure the diversity of user queries would be helpful (XY2i); including human performance as a reference would add clarity (2zSS).
2. **questions regarding the methodological details**: More information is necessary to verify the accuracy of the Table 1 (XY2i); Providing quantitative details on dataset construction—such as time taken for each step and the actual extent of manual effort involved—would be nice (2zSS); why the authors evaluate based on the ReACT framework, and it would be good if they could provide additional evaluation/analysis on other frameworks, such as CodeAct (AydN);
3. **suggestions to conduct additional experiments**: Consider comparisons with more specifically models like AgentLM and xLAM could provide more insights (XY2i); analyse the performance of GPT-4o with the structured outputs. (vqjw).

There are also some suggestions regarding certain phrasing and citations. For more details, we kindly refer you to the specific responses to each reviewer.

---

### Meta-Review · Area_Chair_AT3Z · 2024-12-19

**Metareview:**

The paper "ShortcutsBench: A Large-Scale Real-world Benchmark for API-based Agents" introduces a novel benchmark dataset, ShortcutsBench, designed to evaluate the capabilities of API-based agents in handling real-world complex tasks. The benchmark leverages real APIs from Apple's iOS Shortcuts app, coupled with refined user queries, human-annotated action sequences, and detailed parameter filling values. The authors evaluate a diverse set of large language models (LLMs), including both open-source (e.g., Qwen-2-70B, LLaMA-3-70B) and closed-source (e.g., Gemini-1.5-Pro, GPT-4o-mini) models, on tasks involving API selection, parameter filling, and recognizing the need for additional input. The findings reveal significant limitations in current API-based agents, particularly in multi-step reasoning and handling complex queries. The paper emphasizes the challenges these agents face in real-world scenarios and provides all datasets, code, and experimental logs openly for reproducibility.

The primary contributions of the paper are:
1. **ShortcutsBench Dataset**: A comprehensive, large-scale benchmark for evaluating API-based agents in real-world scenarios.
2. **Extensive LLM Evaluation**: A detailed analysis of 21 LLMs, including both general and agent-specific models, highlighting their capabilities and limitations.
3. **Insights into Agent Limitations**: Valuable findings on the challenges API-based agents face in complex tasks, particularly in parameter filling and recognizing the need for additional input.

Weaknesses

1. **Dataset Construction Details**: Some reviewers noted a lack of detailed description regarding the dataset construction process, particularly the steps taken to ensure query correctness and diversity. While the authors provided additional explanations and open-sourced the process logs, initial clarity was lacking.

2. **Limited Scope of Evaluation Metrics**: The paper primarily focuses on API selection and parameter filling, with limited emphasis on end-to-end task success rates. This was acknowledged by the authors, who noted plans for future work involving dynamic runtime environments.

3. **Presentation Issues**: Some reviewers criticized the initial presentation, citing repetitive citations and unclear phrasing. The authors addressed these concerns in the revised version, but initial readability issues may have impacted the overall reception.

4. **Human Validation Absence**: The absence of human performance baselines was noted as a limitation, as it leaves uncertainty about the benchmark's upper bound and the solvability of tasks. While the authors explained the practical challenges, this remains a valid concern for validating the benchmark's design.

**Additional Comments On Reviewer Discussion:**

1. **Dataset Construction Details (XY2i, AydN):**
   - **Concern**: Reviewers sought more details on dataset construction, particularly regarding action sequence quality and query verification.
   - **Response**: The authors provided a comprehensive explanation of the dataset construction process, including deduplication, API validity checks, and manual verification steps. They also highlighted efforts to ensure query correctness using GPT-4o and human annotations.

2. **Comparison with Agent-Specific Models (XY2i):**
   - **Concern**: Reviewer XY2i suggested evaluating agent-specific models like AgentLM and xLAM for more comprehensive insights.
   - **Response**: The authors conducted extensive additional experiments with 11 models, including agent-specific models, and provided detailed results and analysis. They highlighted the performance differences between fine-tuned and base models, offering valuable insights into agent capabilities.

3. **Clarity and Presentation (AydN, 2zSS):**
   - **Concern**: Reviewers noted unclear descriptions in Section 3 and excessive citations, impacting readability.
   - **Response**: The authors revised Section 3 for clarity, restructured figures, and streamlined citations to enhance readability. They also highlighted key findings more prominently.

4. **Human Validation Absence (2zSS):**
   - **Concern**: Reviewer 2zSS emphasized the need for human performance baselines to validate task solvability and benchmark upper bounds.
   - **Response**: The authors explained the practical challenges of conducting human validation but committed to exploring it in future work. They also provided evidence of manual verification for data quality.

5. **Scientific Value and Conference Fit (vqjw):**
   - **Concern**: Reviewer vqjw questioned the scientific value of the work and suggested a more suitable venue like ICSE or ASE.
   - **Response**: The authors defended the scientific merit of the benchmark by comparing it to other accepted ICLR papers and clarified the relevance of API evaluation in AI research. They also addressed presentation issues and updated citations.

6. **Evaluation Scope and Metrics (2zSS):**
   - **Concern**: Reviewer 2zSS noted the limited evaluation scope, particularly the lack of end-to-end task success metrics.
   - **Response**: The authors acknowledged this limitation and outlined plans for future work involving dynamic runtime environments to assess end-to-end performance.


The authors' responses effectively addressed most reviewer concerns, demonstrating a strong commitment to improving the manuscript. The additional experiments with agent-specific models and the detailed clarifications on dataset construction were particularly impactful. While some limitations, such as the absence of human validation and limited evaluation metrics, persist, these are acknowledged and mitigated by the authors' transparency and plans for future work. The benchmark's real-world relevance and the comprehensive evaluation of diverse LLMs make it a valuable contribution.

---

### Decision · Program_Chairs · 2025-01-22

Accept (Poster)